# $H_2SO_4$-$H_2O$-$NH_3$ ternary ion-mediated nucleation (TIMN): Kinetic-based model and comparison with CLOUD measurements

Fangqun Yu[1], Alexey B. Nadykto[1, 2], Jason Herb[1], Gan Luo[1], Kirill M. Nazarenko[2], and Lyudmila A. Uvarova[2]

Correspondence to: F. Yu (fyu@albany.edu)

[1] Atmospheric Sciences Research Center, University at Albany, Albany, New York, US

[2] Department of Applied Mathematics, Moscow State Univ. of Technology "Stankin", Russia

**Abstract**. New particle formation (NPF) is known to be an important source of atmospheric particles that impacts air quality, hydrological cycle, and climate. Although laboratory measurements indicate that ammonia enhances NPF, the physio-chemical processes underlying the observed effect of ammonia on NPF are yet to be understood. Here we present the first comprehensive kinetically-based $H_2SO_4$-$H_2O$-$NH_3$ ternary ion-mediated nucleation (TIMN) model that is based on the thermodynamic data derived from both quantum-chemical calculations and laboratory measurements. $NH_3$ was found to reduce nucleation barriers for neutral, positively charged, and negatively charged clusters differently, due to large differences in the binding strength of $NH_3$, $H_2O$, and $H_2SO_4$ to small clusters of different charging states. The model reveals the general favor of nucleation of negative ions, followed by nucleation on positive ions and neutral nucleation, for which higher $NH_3$ concentrations are needed, in excellent agreement with CLOUD measurements. The TIMN model explicitly resolves dependences of nucleation rates on all the key controlling parameters, and captures well the absolute values of nucleation rates as well as the dependence of TIMN rates on concentrations of $NH_3$ and $H_2SO_4$, ionization rates, temperature, and relative humidity observed in the well-controlled CLOUD measurements. The kinetic model offers physio-chemical insights into the ternary nucleation process and provides an accurate approach to calculate TIMN rates under a wide range of atmospheric conditions.

## 1. Introduction

New particle formation (NPF), an important source of particles in the atmosphere, is a dynamic process involving interactions among precursor gas molecules, small clusters, and pre-existing particles (Yu and Turco, 2001; Zhang et al., 2012). $H_2SO_4$ and $H_2O$ are known to play an important role in atmospheric particle formation (e.g., Doyle, 1961). In typical atmospheric conditions, the specie dominating the formation and growth of small clusters is $H_2SO_4$. The contribution of $H_2O$ to the nucleation is related to the hydration of $H_2SO_4$ clusters (or, in the other words, modification of the composition of nucleating clusters) that reduces the $H_2SO_4$ vapor pressure and hence diminishes the evaporation of $H_2SO_4$ from the pre-nucleation clusters. $NH_3$, the most abundant gas-phase base molecule in the atmosphere and a very efficient neutralizer of sulfuric acid solutions, has long been proposed to enhance nucleation in the lower troposphere (Coffman and Hegg, 1995) although it has been well recognized that earlier versions of classical ternary nucleation model (Coffman and Hegg, 1995; Korhonen et al., 1999; Napari et al., 2002) significantly over-predict the effect of ammonia (Yu, 2006a; Merikanto et al., 2007; Zhang et al., 2010).

The impacts of $NH_3$ on NPF have been investigated in a number of laboratory studies (Kim et al., 1998; Ball et al., 1999; Hanson and Eisele, 2002; Benson et al., 2009; Kirkby et al., 2011; Zollner et al., 2012; Froyd and Lovejoy, 2012; Glasoe et al., 2015; Schobesberger et al., 2015; Kurten et al., 2016) including those recently conducted at the European Organization for Nuclear Research (CERN) in the framework of the CLOUD (Cosmics Leaving OUtdoor Droplets) experiment that has provided a unique dataset for quantitatively examining the dependences of ternary $H_2SO_4$-$H_2O$-$NH_3$ nucleation rates on concentrations of $NH_3$ ([$NH_3$]) and $H_2SO_4$ ([$H_2SO_4$]), ionization rate (Q), temperature (T), and relative humidity (RH) (Kirkby et al., 2011; Kurten et al., 2016). The experimental conditions in the CLOUD chamber, a 26.1 $m^3$ stainless steel cylinder, were well controlled, while impacts of potential contaminants were minimized (Schnitzhofer et al., 2014; Duplissy et al., 2016). Based on CLOUD measurements in $H_2SO_4$-$H_2O$-$NH_3$ vapor mixtures, Kirkby et al. (2011) reported that an increase of [$NH_3$] from ~ 0.03 ppb (parts per billion, by volume) to ~ 0.2 ppb can enhance ion-mediated (or induced) nucleation rate by 2-3 orders of magnitude and that the ion-mediated nucleation rate is a factor of 2 to >10 higher than that of neutral nucleation under typical level of contamination by amines. In the presence of ionization, highly polar common atmospheric nucleation precursors such as $H_2SO_4$, $H_2O$, and $NH_3$ molecules tend to cluster around ions; and charged clusters are generally much more stable than their neutral counterparts with enhanced growth rates as a result of dipole-charge interactions (Yu and Turco, 2001).

Despite of various laboratory measurements indicate that ammonia enhances NPF, the physio-chemical processes underlying the observed different effects of ammonia on the formation of

neutral, positively charged and negatively charged clusters (Schobesberger et al., 2015) are yet to be understood. To achieve such an understanding, nucleation model based on the first principles is needed. Such a model is also necessary to extrapolate data obtained in a limited number of experimental conditions to a wide range of atmospheric conditions, where [$NH_3$], [$H_2SO_4$], ionization rates, T, RH and surface areas of preexisting particles vary widely depending on the region, pollution level and season. The present work aims to address these issues by developing a kinetically-based $H_2SO_4$-$H_2O$-$NH_3$ ternary ion-mediated nucleation (TIMN) model that is based on the molecular clustering thermodynamic data. The model predictions are compared with relevant CLOUD measurements and previous studies.

## 2. Kinetic-based $H_2SO_4$-$H_2O$-$NH_3$ ternary ion-mediated nucleation (TIMN) model

### 2.1. Background

Most nucleation models developed in the past for $H_2SO_4$-$H_2O$ binary homogeneous nucleation (e.g., Vehkamäki et al., 2002), $H_2SO_4$-$H_2O$ ion-induced nucleation (e.g., Hamill et al., 1982; Raes et al., 1986; Laakso et al., 2003), and $H_2SO_4$-$H_2O$-$NH_3$ ternary homogeneous nucleation (Coffman and Hegg, 1995; Korhonen et al., 1999; Napari et al., 2002) have been based on the classical approach, which employs capillarity approximation (i.e., assuming that small clusters have same properties as bulk) and calculate nucleation rates according to the free energy change associated with the formation of a "critical embryo". Yu and Turco (1997, 2000, 2001) developed a neutral and charged binary $H_2SO_4$-$H_2O$ nucleation model using a kinetic approach that explicitly treats the complex interactions among small air ions, neutral and charged clusters of various sizes, precursor vapor molecules, and pre-existing aerosols. The formation and evolution of cluster size distributions for positively and negatively charged cluster ions and neutral clusters affected by ionization, recombination, neutralization, condensation, evaporation, coagulation, and scavenging, has been named as ion-mediated nucleation (IMN) (Yu and Turco, 2000). The IMN theory significantly differs from classical ion-induced nucleation (IIN) theory (e.g., Hamill et al., 1982; Raes et al., 1986; Laakso et al., 2003) which is based on a simple modification of the free energy for the formation of a "critical embryo" by including the electrostatic potential energy induced by the embedded charge (i.e., Thomson effect (Thomson, 1888)). The classical approach does not properly account for the kinetic limitation to embryo development, enhanced stability and growth of charged clusters associated with dipole-charge interaction (Nadykto and Yu, 2003; Yu, 2005), and the important contribution of neutral clusters resulting from ion-ion recombination to nucleation (Yu and Turco, 2011). In contrast, these important physical processes are explicitly considered in the kinetic-based IMN model (Yu, 2006b).

Since the beginning of the century, nucleation models based on kinetic approach have also been developed in a number of research groups (Lovejoy et al., 2004; Sorokin et al., 2006; Chen

et al., 2012; Dawson et al., 2012; McGrath et al., 2012). Lovejoy et al. (2004) developed a kinetic ion nucleation model, which explicitly treats the evaporation of small neutral and negatively charged $H_2SO_4$-$H_2O$ clusters. The thermodynamic data used in their model were obtained from measurements of small ion clusters, ab initio calculations, thermodynamic cycle, and some approximations (adjustment of Gibbs free energy for neutral clusters calculated based on liquid droplet model, interpolation, etc.). Lovejoy et al. (2004) didn't consider the nucleation on positive ions. Sorokin et al. (2006) developed an ion-cluster-aerosol kinetic (ICAK) model which uses the thermodynamic data reported in Froyd and Lovejoy (2003a, b) and empirical correction terms proposed by Lovejoy et al. (2004). Sorokin et al. (2006) used the ICAK model to simulate dynamics of neutral and charged $H_2SO_4$-$H_2O$ cluster formation and compared the modeling results with their laboratory measurements. Chen et al. (2012) developed an approach for modeling new particle formation based on a sequence of acid-base reactions, with sulfuric acid evaporation rates (from clusters) estimated empirically based on measurements of neutral molecular clusters taken in Mexico City and Atlanta. Dawson et al. (2012) presented a semi-empirical kinetics model for nucleation of methanesulfonic acid (MSA), amines, and water that explicitly accounted for the sequence of reactions leading to formation of stable particles. The kinetic models of Chen et al. (2012) and Dawson et al. (2012) consider only neutral clusters.

McGrath et al. (2012) developed the Atmospheric Cluster Dynamics Code (ACDC) to model the cluster kinetics by solving the birth–death equations explicitly, with evaporation rate coefficients derived from formation free energies calculated by quantum chemical methods. ACDC is also an acid–base reaction model, with the largest clusters containing 4-5 acid and 4-5 base molecules (no water molecules) (Almeida et al., 2013; Olenius et al., 2013). The ACDC model applied to the $H_2SO_4$-dimethylamine (DMA) system considers 0–4 base molecules and 0–4 sulfuric acid molecules (Almeida et al., 2013). Olenius et al. (2013) applied the ACDC model to simulate the steady-state concentrations and kinetics of neutral, and negatively and positively charged clusters containing up to 5 $H_2SO_4$ and 5 $NH_3$ molecules. In ACDC, the nucleation rate is calculated as the rate of clusters growing larger than the upper bounds of the simulated system (i.e., clusters containing 4 or 5 $H_2SO_4$ molecules) (Kurten et al., 2016) and thus may over-predict nucleation rates when critical clusters contain more than 5 $H_2SO_4$ molecules. All clusters simulated by the ACDC model do not contain $H_2O$ molecules and the effect of relative humidity (RH) on nucleation thermochemistry is neglected.

The kinetic IMN model developed by Yu and Turco (1997, 2001) explicitly simulates the dynamics of neutral, positively charged, and negatively charged clusters, based on a discrete-sectional bin structure that covers the clusters containing 0, 1, 2, …, 15, … $H_2SO_4$ molecules to particles containing thousands of $H_2SO_4$ (and $H_2O$) molecules. In the first version of the kinetic IMN model (Yu and Turco, 1997, 2001), due to the lack of thermodynamic data for the small

clusters, the compositions of neutral and charged clusters were assumed to be the same and the evaporation of small clusters was accounted for using a simple adjustment to the condensation accommodation coefficients. Yu (2006b) developed a second-generation IMN model which incorporated newer thermodynamic data (Froyd, 2002; Wilhelm et al., 2004) and physical algorithms (Froyd, 2002; Wilhelm et al., 2004) and explicitly treated the evaporation of neutral and charged clusters. Yu (2007) further improved the IMN model by using two independent measurements (Marti et al., 1997; Hanson and Eisele, 2000) to constrain monomer hydration in the $H_2SO_4$-$H_2O$ system and by incorporating experimentally determined energetics of small neutral $H_2SO_4$-$H_2O$ clusters that became available then (Hanson and Lovejoy, 2006; Kazil et al., 2007). The first and second generations of the IMN model were developed for the $H_2SO_4$-$H_2O$ binary system, although the possible effects of ternary species such as the impact of $NH_3$ on the stability of both neutral and charged pre-nucleation clusters have been pointed out in these previous studies (Yu and Turco, 2001; Yu, 2006b). The present work extends the previous versions of the IMN model in binary $H_2SO_4$-$H_2O$ system to ternary $H_2SO_4$-$H_2O$-$NH_3$ system, as described below. The thermodynamic data sets used for binary clusters were also updated.

## 2.2. Model representation of kinetic ternary nucleation processes

Figure 1 schematically illustrates the evolution of charged and neutral clusters/droplets explicitly simulated in the kinetic $H_2SO_4$-$H_2O$-$NH_3$ TIMN model. Here, $H_2SO_4$ (S) is the key atmospheric nucleation precursor driving the TIMN process while ions, $H_2O$ (W), and $NH_3$ (A) stabilize the $H_2SO_4$ clusters and enhance in this way $H_2SO_4$ nucleation rates. Ions also enhance cluster formation rates due to the interaction with polar nucleating species leading to enhanced collision cross sections (Nadykto and Yu, 2003). The airborne ions are generated by galactic cosmic rays (GCRs) or produced by radioactive emanations, lightning, corona discharge, combustion and other ionization sources. The initial negative ions, which are normally assumed to be $NO_3^-$, are converted into $HSO_4^-$ core ions (i.e., $S^-$) and, then, to larger $H_2SO_4$ clusters in the presence of gaseous $H_2SO_4$. The initial positive ions $H^+W_w$ are converted into $H^+A_{1-2}W_w$ in the presence of $NH_3$, $H^+S_sW_w$ in the presence of $H_2SO_4$, or $H^+A_aS_sW_w$ in the case, when both $NH_3$ and $H_2SO_4$ are present in the nucleating vapors. Some of the binary $H_2SO_4$-$H_2O$ clusters, both neutral and charged, transform into ternary ones by taking up $NH_3$ vapors. The molar fraction of ternary clusters in nucleating vapors depends on [$NH_3$], the binding strength of $NH_3$ to binary and ternary pre-nucleation clusters, cluster composition, and ambient conditions such as T and RH.

Similar to the kinetic binary IMN (BIMN) model (Yu, 2006b), the kinetic TIMN model employs a discrete-sectional bin structure to represent clusters/particles. The bin index $i$ represent the amount of core component (i.e., $H_2SO_4$). For small clusters ($i \le i_d = 30$ in this study), $i$ is the number of $H_2SO_4$ molecules in the cluster (i.e., $i = s$) and the core volume of $i^{th}$ bin $v_i = i \times v_1$, where $v_1$ is the volume of one $H_2SO_4$ molecule. When $i > i_d$, $v_i = VRAT_i \times v_{i-1}$, where $VRAT_i$ is the volume ratio of $i^{th}$ bin to $(i-1)^{th}$ bin. The discrete-sectional bin structure enables the model to cover a wide range of sizes of nucleating clusters/particles with the highest possible size resolution for small clusters (Yu, 2006b). For clusters with a given bin $i$, the associated amounts of water and $NH_3$ and thus the effective radius of each ternary cluster are calculated based on the equilibrium of clusters/particles with the water vapor and/or ammonia, as described in later sections.

The evolution of positive, negative, and neutral clusters due to the simultaneous condensation, evaporation, recombination, coagulation, and other loss processes, is described by the following differential equations obtained by the modification of those describing for the evolution of binary $H_2SO_4$-$H_2O$ system (Yu, 2006b):

$$\frac{\partial N_0^+}{\partial t} = Q + \gamma_1^+ N_1^+ - N_0^+ \left( \sum_{j=1}^{i_{max}} \beta_{i,j}^+ N_j^0 + \sum_{j=0}^{i_{max}} \eta_{i,j}^+ N_j^+ + \sum_{j=0}^{i_{max}} \alpha_{0,j}^{+,-} N_j^- \right) - N_0^+ L_0^+ \tag{1}$$

$$\frac{\partial N_0^-}{\partial t} = Q + \gamma_1^- N_1^- - N_0^- \left( \sum_{j=1}^{i_{max}} \beta_{i,j}^- N_j^0 + \sum_{j=0}^{i_{max}} \eta_{i,j}^- N_j^- + \sum_{j=0}^{i_{max}} \alpha_{0,j}^{-,+} N_j^+ \right) - N_0^- L_0^- \tag{2}$$

$$\begin{aligned}\frac{\partial N_1^0}{\partial t} = {} & P_{H2SO4} + \sum_{j=2}^{i_{max}} \delta_{j,2}\, \gamma_j^0\, N_j^0 + \sum_{j=1}^{i_{max}} (\gamma_j^+\, N_j^+ + \gamma_j^-\, N_j^-) \\ & - N_1^0 \left( \sum_{j=1}^{i_{max}} (1 - f_{1,j,1}) \beta_{1,j}^0 N_j^0 + \sum_{j=0}^{i_{max}} (\beta_{j,1}^+ N_j^+ + \beta_{j,1}^- N_j^-) \right) - N_1^0 L_1^0 \end{aligned} \tag{3}$$

$$\begin{aligned}\frac{\partial N_i^+ (i \ge 1)}{\partial t} = {} & g_{i+1,i}\, \gamma_{i+1}^+\, N_{i+1}^+ - g_{i,i-1}\, \gamma_i^+\, N_i^+ + \sum_{j=0}^{i-1} \sum_{k=1}^{i} \frac{v_j}{v_i} f_{j,k,i} \beta_{j,k}^+ N_j^+ N_k^0 + \sum_{j=0}^{i-1} \sum_{k=0}^{i} \frac{v_j}{v_i} f_{j,k,i} \eta_{j,k}^+ N_j^+ N_k^+ \\ & + \sum_{j=0}^{i} \sum_{k=1}^{i} \frac{v_k}{v_i} f_{j,k,i} \beta_{j,k}^+ N_j^+ N_k^0 - N_i^+ \left( \sum_{j=1}^{i_{max}} (1 - f_{i,j,i}) \beta_{i,j}^+ N_j^0 + \sum_{j=0}^{i_{max}} (1 - f_{i,j,i}) \eta_{i,j}^+ N_j^+ + \sum_{j=0}^{i_{max}} \alpha_{i,j}^{+,-} N_j^- \right) - N_i^+ L_i^+ \end{aligned} \tag{4}$$

$$\begin{aligned}\frac{\partial N_i^- (i \ge 1)}{\partial t} = {} & g_{i+1,i}\, \gamma_{i+1}^-\, N_{i+1}^- - g_{i,i-1}\, \gamma_i^-\, N_i^- + \sum_{j=0}^{i-1} \sum_{k=1}^{i} \frac{v_j}{v_i} f_{j,k,i} \beta_{j,k}^- N_j^- N_k^0 + \sum_{j=0}^{i-1} \sum_{k=0}^{i} \frac{v_j}{v_i} f_{j,k,i} \eta_{j,k}^- N_j^- N_k^- \\ & + \sum_{j=0}^{i} \sum_{k=1}^{i} \frac{v_k}{v_i} f_{j,k,i} \beta_{j,k}^- N_j^- N_k^0 - N_i^- \left( \sum_{j=1}^{i_{max}} (1 - f_{i,j,i}) \beta_{i,j}^- N_j^0 + \sum_{j=0}^{i_{max}} (1 - f_{i,j,i}) \eta_{i,j}^- N_j^- + \sum_{j=0}^{i_{max}} \alpha_{i,j}^{-,+} N_j^+ \right) - N_i^- L_i^- \end{aligned} \tag{5}$$

$$\frac{\partial N_i^0(i \geq 2)}{\partial t} = g_{i+1,i}\, \gamma_{i+1}^0 N_{i+1}^0 - g_{i,i-1}\, \gamma_i^0\, N_i^0 + \sum_{j=1}^{i}\sum_{k=1}^{i-1} \frac{v_k}{v_i} f_{j,k,i} \beta_{j,k}^0 N_j^0 N_k^0$$
$$+ \sum_{j=0}^{i}\sum_{k=0}^{i} f_{j,k,i} \alpha_{j,k}^{+,-} \left(\frac{v_k}{v_i} N_j^+ N_k^- + \frac{v_j}{v_i} N_j^+ N_k^-\right) - N_i^0 \left( \sum_{j=1}^{i_{\max}} (1 - f_{i,j,i}) \beta_{i,j}^0 N_j^0 + \sum_{j=0}^{i_{\max}} (\beta_{j,i}^+ N_j^+ + \beta_{j,i}^- N_j^-) \right) - N_i^0 L_i^0 \quad (6)$$

In Eqs. (1-6), the superscripts "+", "-", and "0" refer to positive, negative, and neutral clusters, respectively, while subscripts *i*, *j*, *k* represent the bin indexes. $N_0^{+,-}$ and $Q$ are the concentration of initial ions not containing $H_2SO_4$ and the ionization rate, respectively. $N_i$ is the total number concentration ($cm^{-3}$) of all cluster/particles (binary + ternary) in the bin *i*. For small clusters ($i \leq i_d$), $N_i$ is the number concentration ($cm^{-3}$) of all clusters containing *i* $H_2SO_4$ molecules. For example, $N_1^0$ is the total concentration of binary and ternary neutral clusters containing one $H_2SO_4$ molecules. $P_{H2SO4}$ is the production rate of neutral $H_2SO_4$ molecules. $L_i^{+,-,0}$ is the loss rate due to scavenging by pre-existing particles, and wall and dilution losses in the laboratory chamber studies (Kirkby et al., 2011; Olenius et al., 2013; Kurten et al., 2016). $f_{j,k,i}$ is the volume fraction of intermediate particles (volume = $v_j + v_k$) partitioned into bin *i* with respect to the core component – $H_2SO_4$, as defined in Jacobson et al. (1994). $g_{i+1,i} = v_1/(v_{i+1} - v_i)$ is the volume fraction of intermediate particles of volume ($v_{i+1} - v_1$) partitioned into bin *i*. $\delta_{j,2}$ =2 at $j$=2 and $\delta_{j,2}$ =1 at $j \neq 2$. $\gamma_i^+$, $\gamma_i^-$, and $\gamma_i^0$ are the mean (or effective) cluster evaporation coefficients for positive, negative and neutral clusters in bin *i*, respectively. $\beta_{i,j}^+$, $\beta_{i,j}^-$, $\beta_{i,j}^0$ are the coagulation kernels for the neutral clusters/particles in bin *j* interacting with positive, negative, and neutral clusters/particles in bin *i*, respectively, which reduce to the condensation coefficients for $H_2SO_4$ monomers at $j$=1. $\eta_{j,k}^+$ and $\eta_{j,k}^-$ are coagulation kernels for clusters/particles of like sign from bin *j* and clusters/particles from bin *k*. $\alpha_{i,j}^{+,-}$ is the recombination coefficient for positive clusters/particles in bin *i* interacting with negative clusters/particles in bin *j*, while $\alpha_{i,j}^{-,+}$ is the recombination

coefficient negative clusters/particles from bin *i* interacting with positively charged clusters/particles from bin *j*.

The methods for calculating *β, γ, η,* and *α* for binary $H_2SO_4$-$H_2O$ clusters have been described in detail in Yu (2006b). Since *β, η,* and *α* depend on the cluster mass (or size) rather than on the cluster composition, schemes for calculating these properties in binary and ternary clusters are identical (Yu, 2006b). In contrast, *γ* is quite sensitive to cluster composition. The evaporation rate coefficient of $H_2SO_4$ molecules from clusters containing *i* $H_2SO_4$ molecules ($\gamma_i$) is largely controlled by the stepwise Gibbs free energy change $\Delta G_{i-1,i}$ of formation of an *i*-mer from an (*i*-1)-mer (Yu, 2007)

$$\gamma_i = \beta_{i-1} N^{o} \exp\left(\frac{\Delta G_{i-1,i}}{RT}\right) \quad (7)$$

$$\Delta G_{k-1,k} = \Delta H^{o}_{k-1,k} - T\Delta S^{o}_{k-1,k} \quad (8)$$

where *R* is the molar gas constant, $N^{o}$ is the number concentration of $H_2SO_4$ at a given T under the reference vapor pressure P of 1 atm. $\Delta H^{o}$ and $\Delta S^{o}$ are enthalpy and entropy changes under the standard conditions (T=298 K, P=1 atm), respectively. The temperature dependence of $\Delta H^{o}$ and $\Delta S^{o}$, which is generally small and typically negligible over the temperature range of interest, was not considered.

$\Delta H$, $\Delta S$ and $\Delta G$ values needed to calculate cluster evaporation rates for the TIMN model can be derived from laboratory measurements and computational quantum chemistry (QC) calculation. Thermochemical properties of neutral and charged binary and ternary clusters obtained using the computational chemical methods and comparisons of computed energies with available experimental data and semi-experimental estimates are given below.

## 2.3. Quantum-chemical studies of neutral and charged binary and ternary clusters

Thermochemical data for small neutral and charged binary $H_2SO_4$-$H_2O$ and ternary $H_2SO_4$-$H_2O$-$NH_3$ clusters has been reported in a number of earlier publications (Bandy and Ianni, 1998; Ianni and Bandy, 1999; Torpo et al., 2007; Nadykto et al., 2008; Herb et al., 2011, 2013; Temelso et al., 2012a, b; DePalma et al., 2012; Ortega et al., 2012; Chon et al., 2014; Husar et al., 2014; Henschel et al., 2014, 2016; Kurten et al., 2015). The PW91PW91/6-311++G(3df,3pd) method,

which is a combination of the Perdue-Wang PW91PW91 density functional with the largest Pople 6-311++G(3df,3pd) basis set, has thoroughly been validated and agrees well with existing experimental data. In earlier studies, this method has been applied to a large variety of atmospherically-relevant clusters (Nadykto et al. 2006, 2007a, b, 2008, 2014, 2015; Torpo et al. 2007; Zhang et al., 2009; Elm et al. 2012; Leverentz et al. 2013; Xu and Zhang, 2012; Xu and Zhang, 2013; Elm et al., 2013; Zhu et al. 2014; Bork et al. 2014; Elm and Mikkelsen, 2014; Peng et al. 2015; Miao et al 2015; Chen et al., 2015; Ma et al., 2016) and has been shown to be well suited to study the ones, as evidenced by a very good agreement of the computed values with measured cluster geometries, vibrational fundamentals, dipole properties and formation Gibbs free energies (Nadykto et al., 2007a, b, 2008, 2014, 2015; Herb et al., 2013; Elm et al., 2012, 2013; Leverentz et al., 2013; Bork et al., 2014) and with high level ab initio results (Temelso et al., 2012a, b; Husar et al., 2012; Bustos et al., 2014).

We have extended the earlier QC studies of binary and ternary clusters to larger sizes. The computations have been carried out using Gaussian 09 suite of programs (Frish et al., 2009). In order to ensure the quality of the conformational search we have carried out a thorough sampling of conformers. We have used both basin hoping algorithm, as implemented in Biovia Materials Studio 8.0, and locally developed sampling code, which creates a "mesh" around the cluster, in which molecules being attached to the cluster are the mesh nodes. Typically, for each cluster of a given chemical composition a thousand to several thousands of isomers have been sampled. We used a three-step optimization procedure, which includes (i) pre-optimization of initial/guess geometries by semi-empirical PM6 method, separation of the most stable isomers located within 15 kcal $mol^{-1}$ of the intermediate global minimum and duplicate removal, followed by (ii) optimization of the selected isomers meeting the aforementioned stability criterion by PW91PW91/CBSB7 method and (iii) the final optimization of the most stable at PW91PW91/CBSB7 level isomers within 5 kcal $mol^{-1}$ of the current global minimum using PW91PW91/6-311++G(3df,3pd) method. Typically, only ~4-30% of initially sampled isomers reach the second (PW91PW91/CBSB7) level, where ~10-40% of isomers optimized with PW91PW91/CBSB7 are selected for the final run. Typically, the number of equilibrium isomers of hydrated clusters is larger than that of unhydrated ones of similar chemical composition. Table 1 shows the numbers of isomers converged at the final PW91PW91/6-311++G(3df,3pd) optimization step for selected clusters and HSG values of the most stable isomers used in the present study. The number of isomers optimized at the PW91PW91/6-311++G(3df,3pd) level of theory varies from case to case, typically being in the range of ~10-200.

The computed stepwise enthalpy, entropy, and Gibbs free energies of cluster formation have been thoroughly evaluated and used to calculate the evaporation rates of $H_2SO_4$ from neutral,

positive and negative charged clusters. A detailed description of QC calculations and the full range of computed properties of binary and ternary clusters will be reported in separate papers.

2.3.1 Positively charged clusters

Table 2 presents the computed stepwise Gibbs free energy changes under standard conditions ($\Delta G^o$) for positive binary and ternary clusters, along with the corresponding experimental data or semi-experimental estimates. Figure 2 shows ΔG associated with the addition of water ($\Delta G^o_{+W}$), ammonia ($\Delta G^o_{+A}$), and sulfuric acid ($\Delta G^o_{+S}$) to binary and ternary clusters as a function of the cluster hydration number *w*.

$H_2O$ has high proton affinity and, thus, $H_2O$ is strongly bonded to all positive ions with low *w*. $\Delta G^o_{+W}$ expectedly becomes less negative and binding of $H_2O$ to binary and ternary clusters weakens due to the screening effect as the hydration number *w* is growing (Fig. 2a). The presence of $NH_3$ in the clusters weakens binding of $H_2O$ to positive ions. For example, $\Delta G^o_{+W}$ for $H^+A_1W_wS_1$ is ~3-4 kcal mol$^{-1}$ less negative than that for $H^+W_wS_1$ at *w*=3-6. The addition of one more $NH_3$ to the clusters to form $H^+A_2W_w$ and $H^+A_2W_wS_1$ further weakens $H_2O$ binding by ~1.5-6 kcal mol$^{-1}$ at *w*=1-3, while exhibiting much smaller impact on hydration free energies at *w*>3. Both the absolute values and trends in $\Delta G^o_{+W}$ derived from calculations are in agreement with the laboratory measurements within the uncertainty range of ~1-2 kcal mol$^{-1}$ for both QC calculations and measurements. This confirms the efficiency and precision of QC methods in calculating thermodynamic data needed for the development of nucleation models.

The proton affinity of $NH_3$ is 204.1 kcal mol$^{-1}$, which is 37.5 kcal mol$^{-1}$ higher than that of $H_2O$ (166.6 kcal mol$^{-1}$) (Jolly, 1991). The hydrated hydronium ions ($H^+W_w$) are easily converted to $H^+A_1W_w$ in the presence of $NH_3$. The binding of $NH_3$ and $H_2O$ molecule to $H^+W_w$ exhibits similar pattern. In particular, binding of $NH_3$ to $H^+W_w$ decreases as *w* is growing, with $\Delta G^o_{+A}$ for $H^+A_1W_w$ ranging from -52.08 kcal mol$^{-1}$ at *w*=1 to -8.32 kcal mol$^{-1}$at *w* = 9. The binding of $NH_3$ to $H^+W_wS_1$ ions is also quite strong, with $\Delta G^o_{+A}$ for $H^+A_1W_wS_1$ ranging from -33.14 kcal mol$^{-1}$ at *w*=1 and to -10.57 kcal mol$^{-1}$ at *w*=6. The addition of the $NH_3$ molecule to $H^+A_1W_w$ (to form $H^+A_2W_w$) is much less favorable thermodynamically than that to $H^+W_w$, with the corresponding $\Delta G^o_{+A}$ being -22 kcal mol$^{-1}$ and -6 kcal mol$^{-1}$ at *w*=2 and *w*=6, respectively. The $\Delta G^o_{+A}$ values for $H^+A_2W_w$ are 3-5 kcal mol$^{-1}$ more negative than the experimental values at *w*=0-1; however, they are pretty close to experimental data at *w*=2-3 (Fig. 2b and Table 2). While it is possible that the QC method overestimates the charge effect on the formation free energies of smallest clusters, the possible overestimation at *w*=0-1 will not affect nucleation calculations because most of $H^+A_2W_w$ in the atmosphere contain more than 2 water molecules (i.e., *w*>2) due to the strong hydration (see Table 2 and Fig. 2a).

Atmospheric Chemistry and Physics
EGU
Discussions

A comparison of QC and semi-experimental estimates of $\Delta G^{o}_{+S}$ values associated with the attachment of $H_2SO_4$ to positive ions shown in Fig. 2c indicates that computed $\Delta G^{o}_{+S}$ values agree well with observations for $H^+W_wS_1$ and $H^+A_1W_wS_1$ but differ by ~2-4 kcal mol$^{-1}$ from semi-experimental values for $H^+A_2W_wS_1$. As seen from Figs. 2a and 2c, the attachment of $NH_3$ to $H^+W_wS_1$ weakens the binding of both $H_2O$ and $H_2SO_4$ to the clusters. This suggests that the attachment of $NH_3$ leads to the evaporation of $H_2SO_4$ and $H_2O$ molecules from the clusters. In other words, $H_2SO_4$ is less stable in $H^+A_1W_wS_1$ than in $H^+W_wS_1$ (Fig. 2c). While this may be taken for the indication that $NH_3$ inhibits nucleation on positive ions at the first look, further calculations show that binding of $NH_3$ to $H^+A_1W_wS_1$ is quite strong (Fig. 2b) and that $H_2SO_4$ in $H^+A_2W_wS_1$ cluster is much more stable than that in $H^+A_1W_wS_1$, with $\Delta G^{o}_{+S}$ being by ~7 kcal mol$^{-1}$ more negative at $w>2$. The $H^+A_2W_wS_1$ cluster can also be formed via the attachment of $H_2SO_4$ to $H^+A_2W_w$. In the presence of sufficient concentrations of $NH_3$, a large fraction of positively charged $H_2SO_4$ monomers exist in the form of $H^+A_2W_wS_1$ and, hence, $NH_3$ enhances nucleation of positive ions. Since positively charged $H_2SO_4$ dimers are expected to contain large number of water molecules, no quantum chemical data for these clusters are available. The CLOUD measurements do indicate that once $H^+A_2W_wS_1$ are formed, they can continue to grow to larger $H^+A_aW_wS_s$ clusters along $a=s+1$ pathway (Schobesberger et al., 2015).

Table 2 and Figure 2 show clearly that the calculated values in most cases agree with measurements within the uncertainty range that justifies the application of QC values in the case, when no reliable experimental data are available.

### 2.3.2 Neutral clusters

Table 3 presents the computed stepwise Gibbs free energy changes for the formation of ternary $S_sA_aW_w$ clusters under standard conditions. The thermodynamic properties of the $S_1A_1$ have been reported in a number of computational studies (e.g., Herb et al., 2011; Kurten et al., 2015; Nadykto and Yu, 2007). However, as pointed out by Kurten et al. (2015), most of these studies, except for Nadykto and Yu (2007), did not consider the impact of $H_2O$ on cluster thermodynamics. We have extended the earlier studies of Nadykto and Yu (2007) and Herb et al. (2011) to larger clusters up to $S_4A_5$ (no hydration) and up to $S_2A_2$ (hydration included). The free energy of binding of $NH_3$ to $H_2SO_4$ (or $H_2SO_4$ to $NH_3$) obtained using our method is -7.77 kcal mol$^{-1}$ that is slightly more negative than values reported by other groups (-6.6 –-7.61 kcal mol$^{-1}$) and within less than 0.5 kcal mol$^{-1}$ of the experimental value of -8.2 kcal mol$^{-1}$ derived from CLOUD measurements (Kurten et al., 2015).

As it may be seen from Table 3, the $NH_3$ binding to $S_{1\text{-}2}W_w$ weakens as $w$ increases. The average $\Delta G^{o}_{+W}$ for $S_1W_w$ formation derived from a combination of laboratory measurements and quantum chemical studies are -3.02, -2.37, and -1.40 kcal mol$^{-1}$ for the first, second, and third

hydration, respectively (Yu, 2007). This indicates that a large fraction of $H_2SO_4$ monomers in the Earth's atmosphere is likely hydrated. Therefore, the decreasing $NH_3$ binding strength to hydrated $H_2SO_4$ monomers implies that RH (and T) will affect the relative abundance of $H_2SO_4$ monomers containing $NH_3$. Currently, no experimental data or observations are available to evaluate the impact of hydration (or RH) on $\Delta G^{o}_{+A}$. Table 3 shows that the presence of $NH_3$ in $H_2SO_4$ clusters suppress hydration and that $\Delta G^{o}_{+W}$ for $S_2A_2$ falls below -2.0 kcal mol$^{-1}$. This is consistent with earlier studies by our group and others showing that large $S_nA_n$ clusters (*n>2*) are not hydrated under typical atmospheric conditions. In the present study, the hydration of neutral $S_nA_n$ clusters at *n>2* is neglected.

The number of $NH_3$ molecules in the cluster (or $H_2SO_4$ to $NH_3$ ratio) significantly affects $\Delta G^{o}_{+S}$ and $\Delta G^{o}_{+A}$ values. For example, $\Delta G^{o}_{+S}$ for $S_3A_a$ clusters increases from -7.08 kcal mol$^{-1}$ to -16.92 kcal mol$^{-1}$ and $\Delta G^{o}_{+A}$ decreases from -16.14 kcal mol$^{-1}$ to -8.93 kcal mol$^{-1}$ as *a* is growing from 1 to 3. For $S_4A_a$ clusters, $\Delta G^{o}_{+S}$ is increasing from -7.48 kcal mol$^{-1}$ to -16.26 kcal mol$^{-1}$ and $\Delta G^{o}_{+A}$ decreases from -17.16 kcal mol$^{-1}$ to -11.34 kcal mol$^{-1}$ as *a* increases from 2 to 4. $\Delta G^{o}_{+A}$ for $S_4A_1$ cluster is by 1.38 kcal mol$^{-1}$ less negative than that for $S_4A_2$. $\Delta G^{o}_{+S}$ for the $S_4A_1$ cluster is also quite low (-4.16 kcal mol$^{-1}$) that might indicate the possible existence of a more stable $S_4A_1$ isomer, which is yet to be identified. In the presence of $NH_3$, the uncertainty in the thermochemistry data for $S_4A_1$ will not significantly affect ternary nucleation rates because most of $S_4$-clusters contain 3 or 4 $NH_3$ molecules.

For the $S_sA_a$ clusters with *s=a*, $\Delta G^{o}_{+A}$ increases as cluster is growing while $\Delta G^{o}_{+S}$ first increases significantly as $S_1A_1$ is converting into $S_2A_2$ and then levels off as $S_2A_2$ is converting into $S_4A_4$. We also observe a significant drop in $\Delta G^{o}_{+A}$ in the case when $NH_3/H_2SO_4$ ratio exceeds 1. This finding is fully consistent with the laboratory measurements showing that growth of neutral $S_sA_a$ clusters follows *s=a* pathway (Schobesberger et al., 2015).

### 2.3.3 Negative ionic clusters

Table 4 shows $\Delta G_{+W}$, $\Delta G_{+A}$, and $\Delta G_{+S}$ needed to form negatively charged clusters under standard conditions, along with available semi-experimental values (Froyd and Lovejoy, 2003). $H_2O$ binding to negatively charged $S^{-}S_s$ clusters significantly strengths with increasing *s*, from $\Delta G^{o}_{+W}$ = -0.61 –-1.83 kcal mol$^{-1}$ at *s*=1-2 to $\Delta G^{o}_{+W}$ = -3.5 kcal mol$^{-1}$ at *w*=1 and -2.25 kcal mol$^{-1}$ at *w*=4 at *s*=4. $\Delta G^{o}_{+W}$ values at *s*=3 and 4 are slightly more negative (by ~ 0.1 – 0.9 kcal mol$^{-1}$ ) than those reported by Froyd and Lovejoy (2003). Just like $H_2O$ binding, $NH_3$ binding to $S^{-}S_s$ at *s*<3 is very weak, with $\Delta G^{o}_{+A}$ ranging from +2.81 kcal mol$^{-1}$ at *s*=0 to -4.85 kcal mol$^{-1}$ at *s*=2. However, it significantly increases as *s* is growing. In particular, at *s*≥3 $\Delta G^{o}_{+A}$ is ranging from -11.89 kcal mol$^{-1}$ for $S^{-}S_3A_1$ to -15.37 kcal mol$^{-1}$ for $S^{-}$ $S_4A_1$. $NH_3$ clearly cannot get into small negative ions. However, it can easily attach to larger negative ions with *s*≥3 that is consistent with CLOUD

measurements (Schobesberger et al., 2015). Since hydration weakens $NH_3$ binding in $S^-S_3A_1W_w$ and $S^-S_4A_1W_w$ clusters, its impacts on the cluster formation and nucleation rates may potentially be important.

In contrast to $H_2O$ and $NH_3$, binding of $H_2SO_4$ to small negative ions ($s$<3) is very strong. These ions are very stable even they contain no $NH_3$ or $H_2O$ molecules. High electron affinity of $H_2SO_4$ molecules results in the high stability of $S^-S_s$ at $s$=1-2. However, the charge effect reduces as $s$ is growing. In particular, $\Delta G^o_{+S}$ of $S^-S_s$ drops from -32.74 kcal mol$^{-1}$ at $s$=1 to -10.58 kcal mol$^{-1}$ and -8.28 kcal mol$^{-1}$ at $s$=3 and 4, respectively. At the same time, $\Delta G^o_{+A}$ increases from 0.08 kcal mol$^{-1}$ ($s$=1) to -11.89 kcal mol$^{-1}$ ($s$=3) and -15.37 kcal mol$^{-1}$ ($s$=4). The hydration of $S^-S_s$ at $s$=3, 4 enhances the strength of $H_2SO_4$ binding, especially at $s=4$. $\Delta G^o_{+S}$ values for $S^-S_{3\text{-}4}W_w$ are consistently ~1.5 – 3 kcal mol$^{-1}$ less negative than the corresponding semi-experimental estimates (Table 4). The possible reasons behind the observed systematic difference are yet to be identified and include the use of low-level *ab initio* HF method to compute reaction enthalpies and uncertainties in experimental enthalpies in studies by Froyd and Lovejoy (2003).

$NH_3$ binding to $S^-S_3$ significantly enhances the stability of $H_2SO_4$ in the cluster by ~7 kcal mol$^{-1}$ compared to $\Delta G^o_{+S}$ for the corresponding binary counterpart. The binding of the second $NH_3$ to $S^-S_3A$ to form $S^-S_3A_2$ is much weaker ($\Delta G^o_{+A}$= -7.27 kcal mol$^{-1}$) that that of the first $NH_3$ molecule ($\Delta G^o_{+A}$= -11.89 kcal mol$^{-1}$). This indicates that most of $S^-A_a$ can only contain one $NH_3$ molecule, in a perfect agreement with the laboratory study of Schobesberger et al. (2015). In the case of $S^-S_4$, binding of the first ( $\Delta G^o_{+A}$= -15.37 kcal mol$^{-1}$) and second (and -12.23 kcal mol$^{-1}$) $NH_3$ molecules to the cluster is quite strong, while the attachment of $NH_3$ leads to substantial stabilization of $H_2SO_4$ in the cluster, as evidenced by $\Delta G^o_{+S}$ growing from -8.28 kcal mol$^{-1}$at $a=0$ to -11.76 kcal mol$^{-1}$and -16.71 kcal mol$^{-1}$ at $a=1$ and $a=2$, respectively. The $NH_3$ binding free energy to $S^-S_4A_2$ (to form $S^-S_4A_3$) drops to -7.59 kcal mol$^{-1,}$ indicating, in agreement with the CLOUD measurements (Schobesberger et al., 2015) that most of $S^-S_4$ clusters contain 1 or 2 $NH_3$ molecules.

### 2.4. Nucleation barriers for neutral/charged clusters and size-dependent evaporation rates

Nucleation barriers and cluster evaporation rates are critically important for calculations of nucleation rates. This section describes the methods employed to calculate the evaporation rates of nucleating clusters of variable sizes and compositions (i.e., $\gamma$ in Eqs. 1-6) in the TIMN model.

#### 2.4.1 Equilibrium distributions of small binary and ternary clusters

In the atmosphere, [$H_2O$] is much higher than [$H_2SO_4$] and, thus, $H_2SO_4$ clusters/particles are always in equilibrium with water vapor (Yu, 2007). In the lower troposphere, where most of the nucleation events were observed, [$H_2SO_4$] is typically at sub-ppt to ppt level, while [$NH_3$] is in the

range of sub-ppb to ppb levels (note that, in what follows, all references to vapor mixing ratios – parts per billion and parts per trillion – are by volume). This means that small ternary clusters can be considered to be in equilibrium with $H_2O$ and $NH_3$ vapors. Like the previous BIMN model derived assuming equilibrium of binary clusters with water vapor, the present TIMN model treats small clusters containing a given number of $H_2SO_4$ molecules as being in equilibrium with both $H_2O$ and $NH_3$. Their relative concentrations are calculated using the thermodynamic data shown in Tables 1-4.

Figure 3 shows the relative abundance (or molar fractions) of small positive, negative, and neutral clusters ($f_{s,a,w}^{+,-,0}$) containing a given number of $H_2SO_4$ molecules at the ambient temperature of 292 K and three different combinations of RH and $[NH_3]$ values. As a result of relative instability of $H_2SO_4$ in $H^+A_1W_wS_1$ compared to $H^+W_wS_1$ or $H^+A_2W_wS_1$ (Fig. 2c), most of positive ions with one $H_2SO_4$ molecule exist in the form of either as $H^+W_wS_1$ or $H^+A_2W_wS_1$ (i.e, containing either zero or two $NH_3$ molecules, Fig. 3a). When $[NH_3]$=0.3 ppb (with T=292 K), most of the positive ions containing one $H_2SO_4$ molecule do not contain $NH_3$ and their composition is dominated by $H^+W_wS_1$ ($\overline{w}$=~7). At the given T and $[NH_3]$=0.3 ppb, around 17% of positive ions with one $H_2SO_4$ molecule contain two $NH_3$ molecules at RH=38%. The fraction of positive ions containing one $H_2SO_4$ and two $NH_3$ molecules decreases to 0.9%, when RH = 90%. At T=292 K and RH=38%, the increase in $[NH_3]$ by a factor of 10 to 3 ppb leads to the domination of $H^+A_2W_wS_1$ (~95%) in the composition of positively charged $H_2SO_4$ monomers. As expected, the composition of positive ions and their contribution to nucleation depends on T, RH, and $[NH_3]$. The incorporation of the quantum chemical and experimental clustering thermodynamics in the framework of the kinetic nucleation model enables us to study all these dependencies.

As a result of very weak binding of $H_2O$ and $NH_3$ to small negative ions (Table 4), nearly all negatively charged clusters with s=0-1 do not contain water and ammonia (not shown). In the case, when *s* is growing to 2, all $S^-S_2A_aW_w$ clusters still do not contain $NH_3$ (i.e., a=0), while only 20-40% of them contain one water molecule (*w*=1) (Fig. 3b). As *s* further increases to 3, $NH_3$ begins to get into some of the negatively charged ions. The fraction of $S^-S_3A_aW_w$ clusters containing one $NH_3$ molecule is 9% at RH=38% and $[NH_3]$=0.3ppb, 3% at RH=90% and $[NH_3]$=0.3 ppb, and 50% at RH=38% and $[NH_3]$=3 ppb. Most of $S^-S_3W_w$ clusters are hydrated while the fraction of $S^-S_3A_aW_w$ clusters containing two $NH_3$ molecules at these ambient conditions is negligible. The fraction of negative cluster ions containing two $NH_3$ molecules becomes significant at *s*=4 (Fig. 3b) and increases from 28% at $[NH_3]$=0.3 ppb to 80% at $[NH_3]$=3 ppb at RH=38%. At $[NH_3]$ =0.3 ppb, the increase in RH from 38% to 90% reduces the fraction of $NH_3$ containing $S^-S_3A_aW_w$ clusters (i.e, *a*>=1) from to 95% to 70%, demonstrating a significant impact of RH on cluster compositions and emphasizing the importance of accounting for the RH in calculations of ternary nucleation rates.

The equilibrium distributions of neutral clusters are presented in Fig. 3c ($H_2SO_4$ monomers and dimers) and Fig. 3d ($H_2SO_4$ trimers and tetramers). Hydration is accounted for in the case of monomers and dimers and not included, due to lack of thermodynamic data, in calculations for trimers and tetramers. Based on the thermodynamic data shown in Table 3, the dominant fraction of neutral monomers is hydrated (79% at RH=38% and 94% at RH=90%) while the fraction of monomers containing $NH_3$ is negligible (0.02% at [$NH_3$]=0.3 ppb and 0.2% at [$NH_3$]=3 ppb, RH=38%). As a result of the growing binding strength of $NH_3$ with the cluster size (Table 3), the fraction of neutral sulfuric acid dimers containing one $NH_3$ molecule reaches 18% at [$NH_3$]=0.3 ppb and 69% at [$NH_3$]=3 ppb when T=292 K and RH=38%. In the case of $H_2SO_4$ trimers and tetramers, data shown in Figure3d are limited to the relative abundance of unhydrated clusters only. Under the given conditions, most of trimers contain two $NH_3$ molecules while most tetramers contain 3 $NH_3$ molecules. At [$NH_3$]=3 ppb, ~2% of trimers contain three $NH_3$ molecules (i.e., $s$=$a$=3) and 55% of tetramers contain four $NH_3$ molecules (i.e., $s$=$a$=4). As a result of a significant drop of $\Delta G^{o}_{+A}$ in the case, when $a/s$ ratio exceeds one (Table 3), the fraction of neutral clusters with $a$=$s$+1 are negligible. The cluster distributions clearly indicate that small sulfuric acid clusters are still not fully neutralized by $NH_3$ even if [$NH_3$] is at ppb level; and that the degree of neutralization (i.e., a:s ratio) increases with the cluster size.

### 2.4.2 Mean stepwise and accumulative Gibbs free energy change and impact of ammonia

In the TIMN model, the equilibrium distributions are used to calculate number concentrations weighted stepwise Gibbs free energy change for adding one $H_2SO_4$ molecule to form a neutral, positively charged, and negatively charged cluster containing $s$ $H_2SO_4$ molecules ($\overline{\Delta G}_{s-1,s}$):

$$\overline{\Delta G}^{+,-,0}_{s-1,s} = \sum_{a,w} f^{+,-,0}_{s,a,w} \Delta G^{+,-,0}_{s-1,s,a,w} \quad (9)$$

where $f^{+,-,0}_{s,a,w}$ is the equilibrium fraction of a particular cluster within a cluster type as shown in Fig. 3.

In the atmosphere, where substantial nucleation is observed, the sizes of critical clusters are generally small ($s$ < ~5–10) and nucleation rates are largely controlled by the stability (or $\gamma$) of small clusters with $s$ < ~5–10. QC calculations and experimental data on clustering thermodynamics available for clusters of small sizes (Tables 2–4), are critically important as the formation of these small clusters is generally the limiting step for nucleation. Nevertheless, thermodynamics data for larger clusters are also needed to develop a robust nucleation model that can calculate nucleation rates under various conditions. Both measurements and QC calculations (Tables 2–4) show significant effects of charge and charge signs (i.e., positive or negative) on the stability and composition of small clusters. These charge effects decrease quickly as the clusters grow, due to the short-ranged nature of dipole-charge interaction and the quick decrease of

electrical field strength around charged clusters as cluster sizes increase (Yu, 2005). Based on experimental data (Kebarle et al., 1967; Davidson et al., 1977; Wlodek et al., 1980; Holland and Castleman, 1982; Froyd and Lovejoy, 2003), the stepwise ΔG values for clusters decreases exponentially as the cluster sizes increase and approaches to the bulk values when clusters containing more than ~ 8-10 molecules (Yu, 2005). Cluster compositions measured with an atmospheric pressure interface time-of-flight (APi-TOF) mass spectrometer during CLOUD experiments also show that the chemical effect of charge-carrying becomes unimportant when the cluster contains more than 9 $H_2SO_4$ molecules (Schobesberger et al., 2015).

In the present TIMN model, we assume that both neutral and charged clusters have the same composition when $s \geq 10$ and the following extrapolation scheme is used to calculate $\Delta G_{s-1,s}$ for clusters up to $s$=10:

$$\Delta G_{s-1,s} = \Delta G_{s_1-1,s_1} + \frac{\left(\Delta G_{s_2-1,s_2} - \Delta G_{s_1-1,s_1}\right)\left(e^{-sc} - e^{-s_1 c}\right)}{\left(e^{-s_2 c} - e^{-s_1 c}\right)} \tag{11}$$

where $\Delta G_{s_1-1,s_1}$ is the stepwise mean Gibbs free energy change for $H_2SO_4$ addition for a specific type (neutral, positive, or negative) of clusters at $s=s_1$ that can be derived from QC calculation and/or experimental measurements, and $\Delta G_{s_2-1,s_2}$ is the corresponding value for clusters at $s=s_2$ (=10 in the present study) that is calculated in the capillarity approximation accounting for the Kelvin effect. $c$ in Eq. 11 is the exponential coefficient that determines how fast $\Delta G_{s-1,s}$ approaches to bulk values as $s$ increases. In the present study, $c$ is estimated from $\Delta G_{s-1,s}$ at $s$=2 and $s$=3 for neutral binary and ternary cluster for which experimental (Hanson and Lovejoy, 2006; Kazil et al., 2007) or quantum-chemical data (Table 3) are available.

For clusters with $s \geq s_2$, the capillarity approximation is used to calculate $\Delta G_{s-1,s}$ as

$$\Delta G_{s-1,s} = -RT \ln(P/P_s) + \frac{2\sigma v_1 N_A}{r_s} \tag{12}$$

where $P$ is the $H_2SO_4$ vapor pressure and $P_s$ is the $H_2SO_4$ saturation vapor pressure over a flat surface with the same composition as the cluster. σ is the surface tension and $v_1$ is the volume of one $H_2SO_4$ molecule. $r_s$ is the radius of the cluster and $N_A$ is the Avogadro's number.

The scheme to calculate bulk $\Delta G_{s-1,s}$ ($s$≥10) for $H_2SO_4$-$H_2O$ binary clusters has been described in Yu (2007). For ternary nucleation, both experiments (Schobesberger et al., 2015) and QC calculations (Table 4) indicate that the growth of relatively large clusters follows the $s$=$a$ line (i.e, in the composition of ammonia bisulfate). In the present TIMN model, the bulk $\Delta G_{s-1,s}$ values for ternary clusters are calculated based on measured $H_2SO_4$ saturation vapor pressure over ammonia bisulfate from Martin et al. (1997) and surface tension from Hyvarinen et al. (2005).

Figure 4 presents stepwise ($\overline{\Delta G}_{s-1,s}$) and cumulative (total) $\overline{\Delta G}_s$ Gibbs free energy changes associated with the formation of neutral, positively charged, and negatively charged binary and ternary clusters containing $s$ $H_2SO_4$ molecules under the conditions specified in the figure caption. The clusters are assumed to be in equilibrium with water (Yu, 2007) and ammonia (Fig. 3). As seen from Fig. 4, the presence of $NH_3$ reduces the mean $\overline{\Delta G}_{s-1,s}$ for larger clusters, which can be treated as the bulk binary $H_2SO_4$-$H_2O$ solution (Schobesberger et al., 2015), by ~ 3 kcal mol$^{-1}$, consistent with the laboratory measurements (Marti et al., 1997) indicating a substantial reduction in the $H_2SO_4$ vapor pressure over ternary solutions. The comparison also shows that the influence of $NH_3$ on $\overline{\Delta G}_{s-1,s}$ of small clusters ($s$≤ ~4) is much lower than that on larger ones and bulk solutions. For example, at [$NH_3$]=0.3 ppb, the differences in $\overline{\Delta G}_{s-1,s}$ between binary and ternary positive ions with $s$=1 and neutral clusters with $s$=2 are only 0.45 kcal mol $^{-1}$ and ~ 1 kcal mol $^{-1}$, respectively. In the case of negative ions, zero and 0.27−0.45 kcal mol$^{-1}$ differences at $s$≤ 2 and $s$=3-4, respectively, were observed. The reduced effect of ammonia on smaller clusters is explained (Tables 2-S4) by ammonia's weaker bonding to smaller clusters than to larger ones, which in turn yields lower average $NH_3$ to $H_2SO_4$ ratios (Fig. 3).

As seen from Fig. 4, bonding of $H_2SO_4$ to small negatively charged clusters ($s$<3) is much stronger than that to neutrals and positive ions. As a result, at $s$<3 the formation of negatively charged clusters is barrierless ($\overline{\Delta G}_{s-1,s}$ <0). $\overline{\Delta G}_{s-1,s}$ (Fig. 4a), and with growing $s$ first increases and then decreases, reaching the maximum barrier values at $s$ = ~ 3 - 6. The effect of $NH_3$ on negative ions becomes important at $s$≥~4, when bonding between the clusters and $NH_3$ becomes strong enough to contaminate a large fraction of binary clusters with ammonia (Fig. 3). In contrast, the impact of $NH_3$ on neutral dimers and positively charged monomers of $H_2SO_4$, as well as on

$\overline{\Delta G}_{s-1,s}$ for both positively charged and neutral clusters, monotonically decreases for all *s*, including $s \leq 5$.

$\overline{\Delta G}_{s-1,s}$ for charged and neutral clusters converge into the bulk values at *s*=~10, when impact of the chemical identity of the core ion on the cluster composition becomes diffuse (Schobesberger et al., 2015) and when the contribution of the electrostatic effect to $\overline{\Delta G}_{s-1,s}$ becomes less than ~ 0.5 kcal mol$^{-1}$. The comparison of cumulative (total) $\overline{\Delta G}_{s}$ (Fig. 4b) indicates the lowest nucleation barrier for the case of negative ions, followed by positive ions and neutrals. The barrierless formation of clusters with *s* ranging from 1 to 3 substantially reduces the nucleation barrier for negatively charged ions and facilitates their nucleation. The presence of 0.3 ppb of $NH_3$ lowers the nucleation barrier for negative, positive and neutral clusters from ~17, 24 and 38 kcal mol$^{-1}$ to 2, 7 and 16 kcal mol$^{-1}$, respectively. A relatively low nucleation barrier for charged ternary clusters is explained by the simultaneous effect of ionization and $NH_3$ which also reduces the size of the critical cluster (*s**).

It is important to note that the size of the critical cluster, commonly used to "measure" the activity of nucleation agents in the classical nucleation theory (Coffman and Hegg, 1995; Korhonen et al., 1999; Vehkamäki et al., 2002; Napari et al., 2002; Hamill et al., 1982) is no longer a valid indicator, when charged molecular clusters and small nanoparticles are considered. As seen from Fig. 4, positively charged ternary critical clusters (*s**=3-4) are smaller than the corresponding negatively charged ones (*s**=4-5); however, the nucleation barrier for ternary positive clusters under the condition is more than three times higher than that for ternary negatives ones.

### 2.4.3 Size- and composition- dependent $H_2SO_4$ evaporation rates

As we mentioned earlier, $H_2SO_4$ is the key atmospheric nucleation precursor driving the formation and growth of clusters in the ternary $H_2SO_4$-$H_2O$-$NH_3$ system while ions, $H_2O$, and $NH_3$ act to stabilize the $H_2SO_4$ clusters. The clustering thermodynamic data derived from QC calculations and measurements (Section 2.3) are used to constrain size- and composition-dependent evaporation rates of $H_2SO_4$ which are critically important. Similar to $\overline{\Delta G}_{s-1,s}$ , average or effective rates of $H_2SO_4$ molecule evaporation from positively charged, negatively charged, and neutral clusters containing *s* $H_2SO_4$ molecules ($\bar{\gamma}_s^{+,-,0}$) are calculated as:

$$\bar{\gamma}_s^{+,-,0} = \sum_{a,w} f_{s,a,w}^{+,-,0} \gamma_{s,a,w}^{+,-,0} \quad (10)$$

where $\gamma_{s,a,w}^{+,-,0}$ is the $H_2SO_4$ evaporation coefficient from a particular cluster within a cluster type as shown in Fig. 3, which can be calculated based on Eq. (7) with $\Delta G_{+S}^{0}$ from Tables 2-4.

Figure 5 gives the number concentration weighted mean evaporation rate ($\bar{\gamma}$) of an $H_2SO_4$ molecule from these clusters under the conditions corresponding to Fig. 4. The shapes of $\bar{\gamma}$ curves are similar to those of $\overline{\Delta G}_{s-1,s}$ (Fig. 4a) as $\bar{\gamma}$ values are largely controlled by $\overline{\Delta G}_{s-1,s}$ (Eq. 7). The presence of ammonia, as expected, significantly reduces the vapor pressure of $H_2SO_4$ over bulk aerosol (Marti et al., 1997), and, hence, the $H_2SO_4$ evaporation rate. The evaporation rates of both neutral and positive clusters decrease as *s* increases, and the positive clusters are uniformly more stable than corresponding neutral clusters. $\bar{\gamma}$ for negative ions first increases and then decreases as *s* increases, peaking around *s*= ~3 - 6. The presence of $NH_3$ reduces the evaporation rates of larger clusters by more than two orders of magnitude and the effect decreases for smaller clusters, as the binding of $NH_3$ to small neutral and charged clusters are weaker compared to that for larger clusters (Fig. 4). [$NH_3$] influences the average $NH_3$:$H_2SO_4$ ratio (Fig. 3) and the evaporation rates of these small clusters. The nucleation rates, limited by formation of small clusters (*s*<~ 5), depend strongly on the stability or evaporation rate of these small clusters and, thus, on [$NH_3$].

## 3. TIMN rates and comparisons with CLOUD measurements

The evolution of cluster/particle size distributions can be obtained by solving the dynamic equations 1-6. Since the concentrations of clusters of all sizes are explicitly predicted, the nucleation rates in the kinetic model can be calculated for any cluster size larger than the critical size of neutral clusters (*i* >*i**) (Yu, 2006b),

$$J_i = J_i^+ + J_i^- + J_i^0 = \beta_{i,1}^+ N_1^0 N_i^+ - \gamma_i^+ N_{i+1}^+ + \beta_{i,1}^- N_1^0 N_i^- - \gamma_i^- N_{i+1}^- + \beta_{i,1}^0 N_1^0 N_i^0 - \gamma_i^0 N_{i+1}^0 \qquad (13)$$

where $J_i^+$, $J_i^-$, and $J_i^0$ are nucleation rates associated with positive, negative, and neutral clusters containing *i* $H_2SO_4$ molecules. As a result of scavenging by pre-existing particles or wall loss, the steady state $J_i$ decreases as *i* increases. To compare with CLOUD measurements, we calculate nucleation at cluster mobility diameter of 1.7 nm ($J_{1.7}$).

Many practical applications require information on the steady state nucleation rates. For each nucleation case presented in this paper, constant values of [$H_2SO_4$] (i.e., $N_1^0$), [$NH_3$], T, RH, Q, and $L_i^{+,-,0}$ are assumed. The pre-existing particles with fixed surface area or wall loss serve as a sink for all clusters. Under a given condition, cluster distribution and nucleation rate reach steady state after a certain amount of time. We calculate size-dependent coefficients for a given case, and

then solve equations (1-6) to obtain the steady state cluster distribution and nucleation rate, with the approach described in Yu (2006b).

Figure 6 shows a comparison of the model TIMN rates $J_{1.7}$ with CLOUD measurements, as a function of [$NH_3$] under two ionization rates. It should be noted that Dunne et al. (2016) developed a simple empirical parameterization (denoted thereafter as "CLOUDpara") of binary, ternary and ion-induced nucleation rates in CLOUD measurements as a function of [$NH_3$], [$H_2SO_4$], T, and negative ion concentration. The predictions based on CLOUDpara (Dunne et al., 2016) and ACDC (McGrath et al., 2012; Kurten et al., 2016) are also presented in Fig. 6 for comparisons.

Like the CLOUD measurements, the TIMN predictions reveal a complex dependence of $J_{1.7}$ on [$NH_3$], and an analysis of the TIMN results shows this behavior can be explained by the differing responses of negative, positive and neutral clusters to the presence of ammonia (Fig. 4). Under the conditions specified in Fig. 6, nucleation is dominated by negative ions for [$NH_3$] <~0.5 ppb, by both negative and positive ions for [$NH_3$] from ~0.5 ppb to ~10 ppb (with background ionization), or ~20 ppb (with pion-enhanced ionization), and by neutrals at higher [$NH_3$]. According to TIMN, [$NH_3$] of at least 0.6–1 ppb are needed before positive ions contribute significantly to nucleation rates – in good agreement with the threshold found in the CLOUD experiments (Kirkby et al., 2011; Schobesberger et al., 2015). TIMN simulations also extend CLOUD data at [$NH_3$] of ~1 ppb to include a "zero-sensitivity zone" in the region of 1-10 ppb, followed by a region of strong sensitivity of $J_{1.7}$ to [$NH_3$] commencing at [$NH_3$] > ~10-20 ppb. The latter zone may have important implications for NPF in heavily polluted regions, including much of India and China, where [$NH_3$] may exceed 10-20 ppb (Behera and Sharma, 2010; Meng et al., 2017). It is noteworthy in Fig. 6 that the dependence of $J_{1.7}$ on [$NH_3$] and Q predicted by the ACDC model (McGrath et al., 2012) and the CLOUD data parameterization (Dunne et al., 2016) deviate substantially from the experimental data as well as the TIMN simulations. The CLOUDpara does not consider impacts of positive ions and such key controlling parameters as RH and surface area of pre-existing particles. Dunne et al. (2016) reported that CLOUDpara is also very sensitive to the approach to parameterize T dependence, showing that the contribution of ternary ion-induced nucleation to NPF below 15 km altitude has grown from 9.6% to 37.5%, after the initial empirical temperature function was replaced with a simpler one.

Figure 7 presents a more detailed comparison of TIMN simulations with CLOUD measurements of $J_{1.7}$ as a function of [$H_2SO_4$], T, and RH. The TIMN model accurately reproduces both the absolute values of $J_{1.7}$ and its dependencies on [$H_2SO_4$], T, and RH, in a wide range of temperatures (T=208 – 292 K) and [$H_2SO_4$] ($5\times10^5$ – $5\times10^8$ $cm^{-3}$). As expected, nucleation rates are very sensitive to [$H_2SO_4$] and T. For example, $J_{1.7}$ increases by three to five orders of magnitude with an increase in [$H_2SO_4$] of a factor of 10, and by roughly one order of magnitude for a temperature decrease of 10 degree, except in cases where the nucleation rate is limited by Q (for

example, $[H_2SO_4]$ =~$10^8$ – $10^9$ $cm^{-3}$ at T=278 K and 292 K, shown in Fig. 7a). The key difference between CLOUDpara and TIMN predictions is that $dlnJ_{1.7}/dln[H_2SO_4]$ ratio predicted by CLOUDpara is nearly constant while TIMN shows that this ratio depends on both $[H_2SO_4]$ and T. The CLOUD measurements taken at T=278 K clearly show (in agreement with the TIMN) that $dlnJ_{1.7}/dln[H_2SO_4]$ is not constant. CLOUDpara overestimates $J_{1.7}$ compared to both measurements and TIMN simulations, except for the case, when T=278 K and $[H_2SO_4]$ ranges from ~$7\times10^6$ to $5\times10^7$ $cm^{-3}$, with deviation of CLOUDpara from experimental data and TIMN growing with the lower temperature.

Both CLOUD measurements and TIMN simulations (Fig. 7b) show an important influence of RH on nucleation rates (which is neglected in both the CLOUDpara and ACDC models). In particular, CLOUD measurements indicate 1-5 order of magnitude rise in $J_{1.7}$ after RH increases from 10% to 70-80% and a stronger effect of RH on nucleation rates at higher temperatures under the conditions shown in Fig. 7b. The RH dependence of $J_{1.7}$ predicted by the TIMN model is consistent with measurements, being slightly weaker than the measured at high RH.

## 4. Summary

A comprehensive kinetically-based $H_2SO_4$-$H_2O$-$NH_3$ ternary ion-mediated nucleation (TIMN) model, constrained with thermodynamic data from quantum-chemical calculations and laboratory measurements, has been developed and used to shed a new light on physio-chemical processes underlying the effect of ammonia on NPF. We show that the stabilizing effect of $NH_3$ grows with the cluster size, and that the reduced effect of ammonia on smaller clusters is caused by weaker bonding that in turn yields lower average $NH_3$ to $H_2SO_4$ ratios. $NH_3$ was found to impact nucleation barriers for neutral, positively charged, and negatively charged clusters differently due to the large difference in the binding energies of $NH_3$, $H_2O$, and $H_2SO_4$ to small clusters of different charging states. The lowest and highest nucleation barriers are observed in the case of negative ions and neutrals, respectively. Therefore, nucleation of negative ions is favorable, followed by nucleation of positive ions and neutrals. Different responses of negative, positive and neutral clusters to ammonia result in a complex dependence of ternary nucleation rates on $[NH_3]$. The TIMN model reproduces both the absolute values of nucleation rates and their dependencies on the key controlling parameters and agrees with the CLOUD measurements much better than other models being tested here over a wide range of ambient conditions encompassing those encountered in the global atmosphere.

The TIMN model developed in the present study may subject to uncertainties associated with the use of experimental and thermodynamic data for pre-nucleation clusters. Further measurements and quantum calculations are needed to reduce the uncertainties. While the TIMN model predicts nucleation rates in a good agreement with the CLOUD measurements, its ability to explain the

NPF events observed in the real atmosphere is yet to be quantified and will be investigated in further studies.

**Acknowledgments.** The authors thank Richard Turco (Distinguished Professor Emeritus, UCLA) for comments that helped to improve the manuscript. This study was supported by NSF under grant 1550816, NASA under grant NNX13AK20G, and NYSERDA under contract 100416. ABN would like to thank the Russian Ministry of Education and Science for its support.

**Data availability.** All relevant data are available in the article, or from the corresponding authors upon request.

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

**Table 1.** Number of isomers successfully converged at 6-311 level for selected clusters, along with the enthalpy (H), entropy (S), and Gibbs free energy (G) of the most stable isomers.

| Cluster Formula | 6-311++ conv. | H | S | G |
|---|---|---|---|---|
| $S_4$ | 56 | -2801.256008 | 179.461 | -2801.341276 |
| $S_4A_1$ | 169 | -2857.820795 | 187.395 | -2857.909833 |
| $S_4A_2$ | 84 | -2914.388489 | 193.997 | -2914.480663 |
| $S_4A_3$ | 68 | -2970.94645 | 209.77 | -2971.046119 |
| $S_4A_4$ | 38 | -3027.500303 | 225.959 | -3027.607663 |
| $S_4A_5$ | 34 | -3084.050337 | 237.758 | -3084.163303 |
| $S^- S_3$ | 97 | -2800.835072 | 168.993 | -2800.915366 |
| $S^- S_3A_1$ | 122 | -2857.389946 | 184.899 | -2857.477797 |
| $S^- S_3A_2$ | 21 | -2913.941409 | 192.489 | -2914.032867 |
| $S^- S_3A_3$ | 13 | -2970.490814 | 195.627 | -2970.583762 |
| $S^- S_4$ | 138 | -3501.162655 | 200.525 | -3501.257931 |
| $S^- S_4A_1$ | 71 | -3557.727072 | 208.015 | -3557.825907 |
| $S^- S_4A_2$ | 22 | -3614.287482 | 213.397 | -3614.388874 |
| $S^- S_4A_3$ | 23 | -3670.836831 | 226.504 | -3670.94445 |
| $S^- S_4A_4$ | 18 | -3727.385956 | 237.152 | -3727.498634 |
| $H^+A_2$ | 16 | -113.413269 | 68.478 | -113.445805 |
| $H^+A_2W_1$ | 42 | -189.845603 | 94.248 | -189.890384 |
| $H^+A_2W_2$ | 56 | -266.276653 | 113.49 | -266.330576 |
| $H^+A_2W_3$ | 63 | -342.706301 | 132.722 | -342.769362 |
| $H^+A_2W_4$ | 114 | -419.133157 | 160.449 | -419.209391 |
| $H^+A_2W_5$ | 116 | -495.567408 | 161.447 | -495.644117 |
| $H^+A_2W_6$ | 70 | -571.994961 | 175.085 | -572.078149 |
| $H^+A_2W_0S_1$ | 40 | -813.745253 | 107.764 | -813.796455 |
| $H^+A_2W_1S_1$ | 173 | -890.181285 | 121.33 | -890.238933 |
| $H^+A_2W_2S_1$ | 103 | -966.618165 | 130.584 | -966.680209 |
| $H^+A_2W_3S_1$ | 169 | -1043.047622 | 154.145 | -1043.120861 |
| $H^+A_2W_4S_1$ | 188 | -1119.476882 | 177.051 | -1119.561004 |
| $H^+A_2W_5S_1$ | 178 | -1195.90253 | 200.029 | -1195.99757 |
| $H^+A_2W_6S_1$ | 85 | -1272.330781 | 215.117 | -1272.43299 |

**Table 2.** QC-based stepwise Gibbs free energy change for the addition of one water ($\Delta G^{o}_{+W}$), ammonia ($\Delta G^{o}_{+A}$), or sulfuric acid ($\Delta G^{o}_{+S}$) molecule to form the given positively charged clusters under standard conditions, and the corresponding experimental data or semi-experimental estimates.

| | $\Delta G^{o}_{+W}$ | | $\Delta G^{o}_{+A}$ | | $\Delta G^{o}_{+S}$ | |
|---|---|---|---|---|---|---|
| | QC | experimental | QC | experimental | QC | experimental |
| $H^+W_1S_1$ | | | | | -28.59 | -24.65 [f] |
| $H^+W_2S_1$ | -15.66 | | | | -15.33 | -13.76 [f] |
| $H^+W_3S_1$ | -9.40 | | | | -10.12 | -11.93 [f] |
| $H^+W_4S_1$ | -7.83 | | | | -9.18 | -9.71 [f] |
| $H^+W_5S_1$ | -6.77 | -5.79 [a] | | | -9.52 | -9.82 [f] |
| $H^+W_6S_1$ | -5.32 | -4.24 [a] | | | -9.70 | -9.94 [f] |
| $H^+W_7S_1$ | -3.18 | -3.28 [a] | | | -9.64 | -9.96 [f] |
| $H^+W_8S_1$ | -2.80 | -2.67 [a] | | | -9.84 | -10.10 [f] |
| $H^+W_9S_1$ | -2.30 | -2.12 [a] | | | -10.24 | -10.86 [f] |
| $H^+A_1W_1$ | -13.47 | -13.01 [b], -11.43 [c] | -52.08 | | | |
| $H^+A_1W_2$ | -9.85 | -7.14 [b], -8.17 [c] | -33.02 | | | |
| $H^+A_1W_3$ | -6.60 | -5.92 [b], -5.88 [c] | -25.01 | | | |
| $H^+A_1W_4$ | -3.50 | -3.94 [b], -4.06 [c] | -19.73 | | | |
| $H^+A_1W_5$ | -2.50 | -2.55 [b],-3.02 [c] | -15.80 | | | |
| $H^+A_1W_6$ | -2.26 | -2.54 [b] | -12.93 | | | |
| $H^+A_1W_7$ | -1.15 | -1.84 [b] | -10.84 | | | |
| $H^+A_1W_8$ | -1.02 | | -9.26 | | | |
| $H^+A_1W_9$ | 0.25 | | -8.32 | | | |
| $H^+A_2$ | | | -22.97 | -18.25 [c] | | |
| $H^+A_2W_1$ | -7.04 | -6.85 [c] | -16.53 | -11.54 [c], -12.75 [d] | | |
| $H^+A_2W_2$ | -4.29 | -5.25 [c] | -10.97 | -9.13 [c], -9.50 [d] | | |
| $H^+A_2W_3$ | -3.41 | -3.70 [c] | -7.78 | -6.83 [c], -7.02 [d] | | |
| $H^+A_2W_4$ | -3.08 | | -7.36 | | | |
| $H^+A_2W_5$ | -1.97 | | -6.82 | | | |
| $H^+A_2W_6$ | -0.42 | | -4.99 | | | |
| $H^+A_1W_1S_1$ | -8.99 | | -33.14 | | -9.65 | -8.3 [d] |
| $H^+A_1W_2S_1$ | -8.11 | | -25.59 | | -7.90 | -7.1 [d] |
| $H^+A_1W_3S_1$ | -6.09 | | -22.28 | | -7.40 | -6.7 [d] |
| $H^+A_1W_4S_1$ | -4.25 | | -18.71 | | -8.15 | -6.9 [d] |
| $H^+A_1W_5S_1$ | -1.92 | | -13.85 | | -7.56 | -7.5 [d] |
| $H^+A_1W_6S_1$ | -2.04 | | -10.57 | | -7.34 | -8.0 [d] |
| $H^+A_2W_0S_1$ | | | -22.09 | -22.14 [e] | -13.35 | -16.8 [d] |

| | | | | |
|---|---|---|---|---|
| $H^+A_2W_1S_1$ | -5.72 | -18.92 | -12.03 | -15.8 [d] |
| $H^+A_2W_2S_1$ | -4.97 | -15.78 | -12.71 | -15.9 [d] |
| $H^+A_2W_3S_1$ | -4.58 | -14.27 | -13.89 | -16.3 [d] |
| $H^+A_2W_4S_1$ | -4.26 | -14.27 | -15.06 | -17.3 [d] |
| $H^+A_2W_5S_1$ | -2.01 | -14.37 | -15.11 | -18.8 [d] |
| $H^+A_2W_6S_1$ | -1.29 | -13.63 | -15.98 | -19.9 [d] |

[a] Froyd and Lovejoy, 2003; [b] Meot-Ner (Mautner) et al., 1984; [c] Payzant et al., 1973; [d] Froyd, 2002; [e] Froyd and Lovejoy, 2012. [f] The $\Delta G^{o}_{+S}$ values given here were calculated based experimental $\Delta G^{o}_{+S}$ values at T=270 K from Froyd and Lovejoy (2003) and ΔS values from quantum calculation.

**Table 3.** Same as Table 2 except for neutral clusters.

| | $\Delta G^o_{+W}$ | | $\Delta G^o_{+A}$ | | $\Delta G^o_{+S}$ | |
|---|---|---|---|---|---|---|
| | QC | experimental | QC | experimental | QC | experimental |
| $S_1A_1$ | | | -7.77 [a] (-7.29 [b], -7.61 [c], -6.60 [d]) | - 8.2 [e] | -7.77 [a] (-7.29 [b], -7.61 [c], -6.60 [d]) | - 8.2 [e] |
| $S_1A_1W_1$ | -1.39 [a] | | -6.88 [a] | | | |
| $S_1A_1W_2$ | -2.30 [a] | | -6.18 [a] | | | |
| $S_1A_1W_3$ | -1.52 [a] | | -5.81 [a] | | | |
| $S_1A_2$ | | | -4.75 | | | |
| $S_1A_2W_1$ | -0.78 | | -4.15 | | | |
| $S_2A_1$ | | | -13.84 [a] | | -11.65 [a] | |
| $S_2A_1W_1$ | -2.31 [a] | | -12.77 | | -12.59 [a] | |
| $S_2A_1W_2$ | -1.21 [a] | | -11.00 | | -11.52 [a] | |
| $S_2A_1W_3$ | -2.04 [a] | | -9.69 | | -12.04 [a] | |
| $S_2A_2$ | | | -8.75 | | -15.65 | |
| $S_2A_2W_1$ | -1.96 | | -8.37 | | -16.83 | |
| $S_2A_2W_2$ | -1.19 | | -8.35 | | -15.49 | |
| $S_2A_2W_3$ | 0.60 | | -5.71 | | -14.42 | |
| $S_2A_3$ | | | -4.19 | | | |
| $S_3A_1$ | | | -16.14 | | -7.08 | |
| $S_3A_2$ | | | -13.84 | | -12.17 | |
| $S_3A_3$ | | | -8.93 | | -16.92 | |
| $S_3A_4$ | | | -7.42 | | | |
| $S_4A_1$ | | | -15.74 | | -4.16 | |
| $S_4A_2$ | | | -17.16 | | -7.48 | |
| $S_4A_3$ | | | -13.79 | | -12.34 | |
| $S_4A_4$ | | | -11.34 | | -16.26 | |
| $S_4A_5$ | | | -7.63 | | | |

[a] Nadykto and Yu, 2007; [b] Torpo et al., 2007; [c] Ortega et al., 2012; [d] Chon et al., 2007; [e] Kurten et al., 2015.

**Table 4.** Same as Table 2 except for negatively charged clusters.

| | $\Delta G^{o}_{+W}$ | | $\Delta G^{o}_{+A}$ | | $\Delta G^{o}_{+S}$ | |
|---|---|---|---|---|---|---|
| | QC | experimental | QC | experimental | QC | experimental |
| $S^- A_1$ | | | 2.81 | | | |
| $S^- S_1W_0$ | | | | | -32.74 | -29.10[a] |
| $S^- S_1W_1$ | -0.61 | | | | -28.12 | |
| $S^- S_1W_2$ | -1.06 | | | | -25.36 | |
| $S^- S_1A_1$ | | | 0.08 | | -35.47 | |
| $S^- S_2W_0$ | | | | | -15.06 | -17.14[a] |
| $S^- S_2W_1$ | -1.83 | | | | -16.28 | |
| $S^- S_2A_1$ | | | -4.85 | | -19.99 | |
| $S^- S_3W_0$ | | | | | -10.58 | -13.28 [a] |
| $S^- S_3W_1$ | -2.92 | -2.73 [a] | | | -11.67 | -14.29 [a] |
| $S^- S_3W_2$ | -2.03 | -1.53 [a] | | | -11.12 | -13.80 [a] |
| $S^- S_3W_3$ | -2.01 | -1.93 [a] | | | -11.52 | -14.72 [a] |
| $S^- S_3W_4$ | -1.73 | | | | | |
| $S^- S_3A_1W_0$ | | | -11.89 | | -17.62 | |
| $S^- S_3A_1W_1$ | 0.52 | | -8.45 | | -14.90 | |
| $S^- S_3A_1W_2$ | 0.39 | | -6.03 | | -13.06 | |
| $S^- S_3A_2$ | | | -7.27 | | -18.36 | |
| $S^- S_3A_3$ | | | -4.66 | | | |
| $S^- S_4W_0$ | | | | | -8.28 | -10.96 [a] |
| $S^- S_4W_1$ | -3.50 | -2.61 [a] | | | -8.86 | -10.71 [a] |
| $S^- S_4W_2$ | -3.17 | -2.79 [a] | | | -9.99 | -12.10 [a] |
| $S^- S_4W_3$ | -2.65 | -2.41 [a] | | | -10.64 | -12.48 [a] |
| $S^- S_4W_4$ | -2.25 | -2.14 [a] | | | -11.16 | -12.77 [a] |
| $S^- S_4A_1W_0$ | | | -15.37 | | -11.76 | |
| $S^- S_4A_1W_1$ | -2.21 | | -14.09 | | -14.49 | |
| $S^- S_4A_1W_2$ | -0.74 | | -11.66 | | -15.62 | |
| $S^- S_4A_2$ | | | -12.23 | | -16.71 | |
| $S^- S_4A_3$ | | | -7.59 | | -19.65 | |
| $S^- S_4A_4$ | | | -6.72 | | | |

[a] Froyd and Lovejoy, 2003.

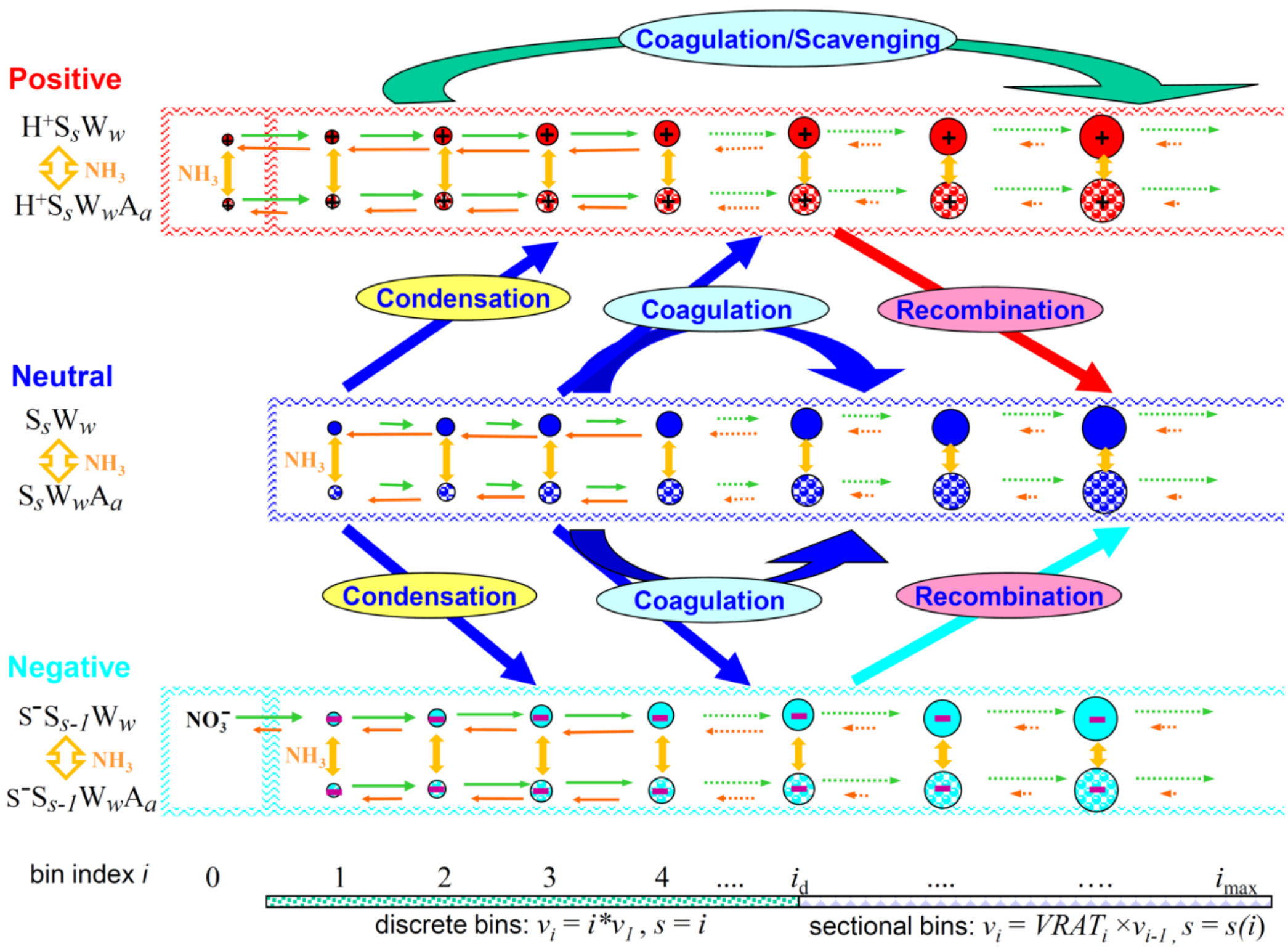

**Figure 1.** Schematic illustration of kinetic processes controlling the evolution of positively charged ( $H^+S_sW_wA_a$ ), neutral ( $S_sW_wA_a$ ), and negatively charged ( $S^-S_{s-1}W_wA_a$ ) clusters/droplets that are explicitly simulated in the ternary ion-mediated nucleation (TIMN) model. Here S, W, and A represent sulfuric acid ($H_2SO_4$), water ($H_2O$), and ammonia ($NH_3$) respectively, while *s*, *w*, and *a* refer to the number of S, W, and A molecules in the clusters/droplets, respectively. The TIMN model has been extended from an earlier version treating binary IMN (BIMN) by adding $NH_3$ into the nucleation system and using a discrete-sectional bin structure to represent the sizes of clusters/particles starting from a single molecule up to background particles larger than a few micrometers.

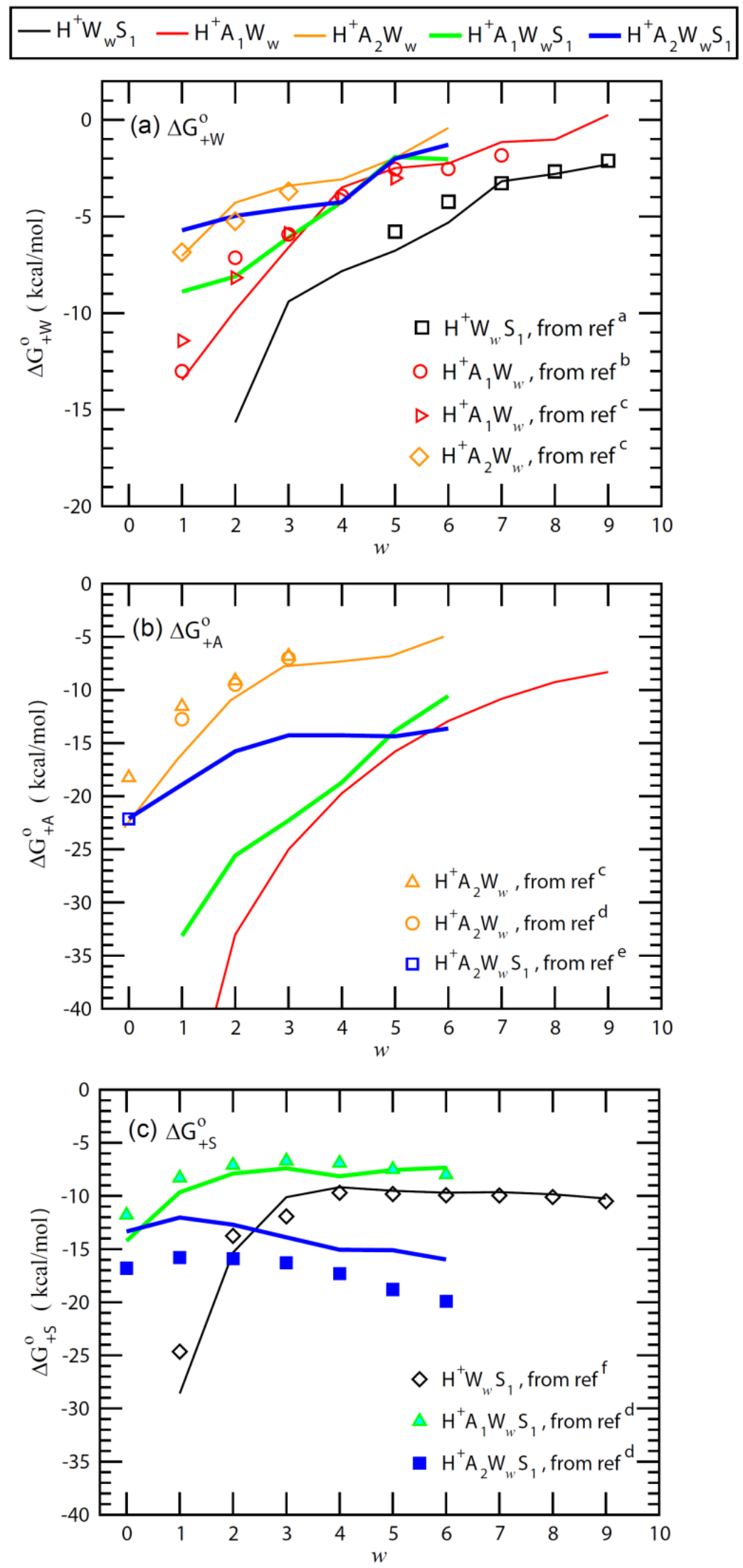

**Figure 2.** Stepwise Gibbs free energy change under standard conditions for the addition of a water ($\Delta G^{o}_{+W}$), ammonia ($\Delta G^{o}_{+A}$), or sulfuric acid ($\Delta G^{o}_{+S}$) molecule to form the given positively charged clusters as a function of the number of water molecules in the clusters (*w*). Lines are QC-based values, and symbols are experimental results or semi-experimental estimates (see notes under Table 2 for the references).

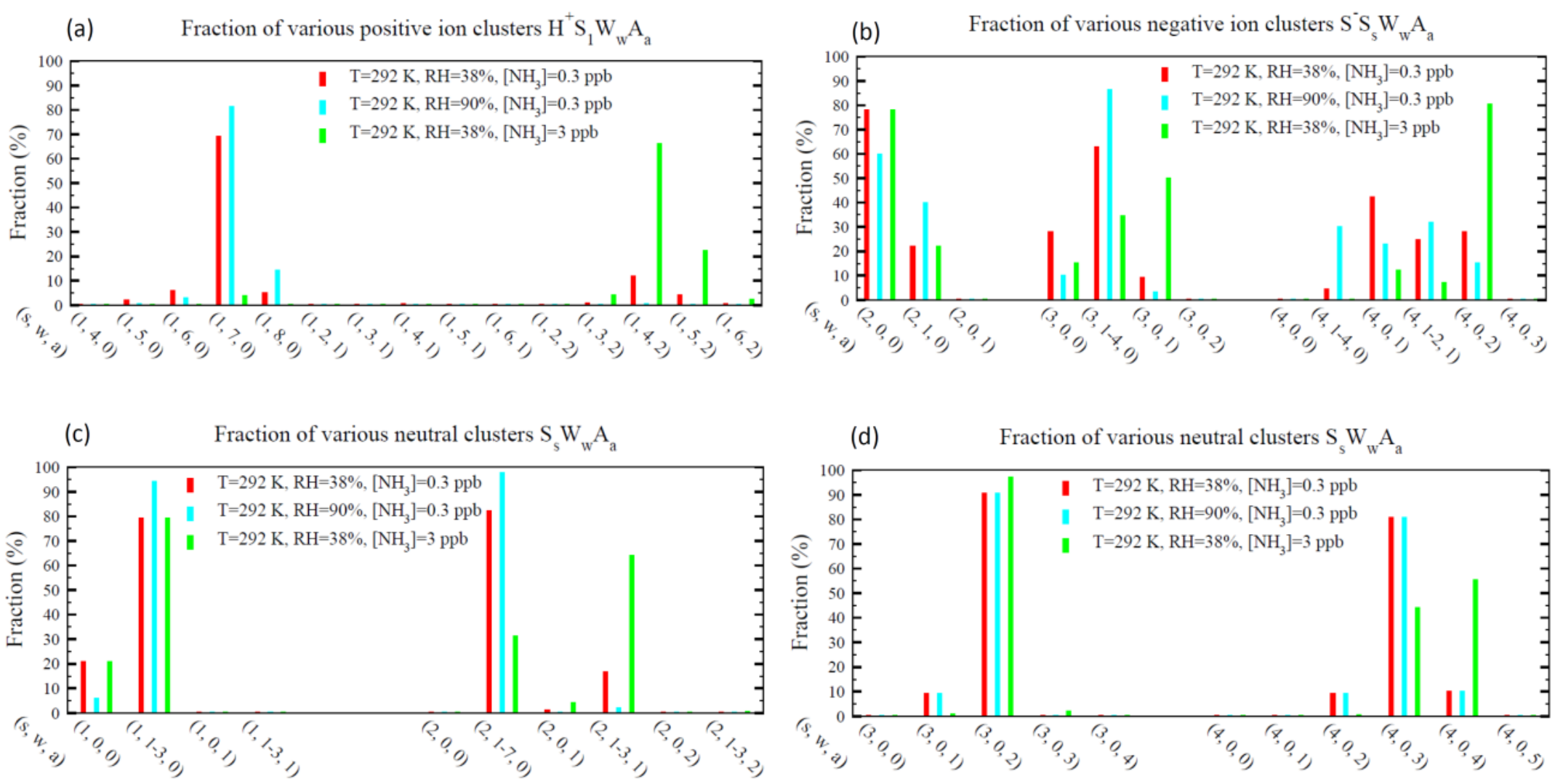

**Figure 3**. Relative abundance (or molar fraction) of small clusters containing a given number of $H_2SO_4$ molecules for positive, negative, and neutral cluster types at a temperature of 292 K and three different combinations of RHs (38% and 90%) and [$NH_3$] (0.3 and 3 ppb).

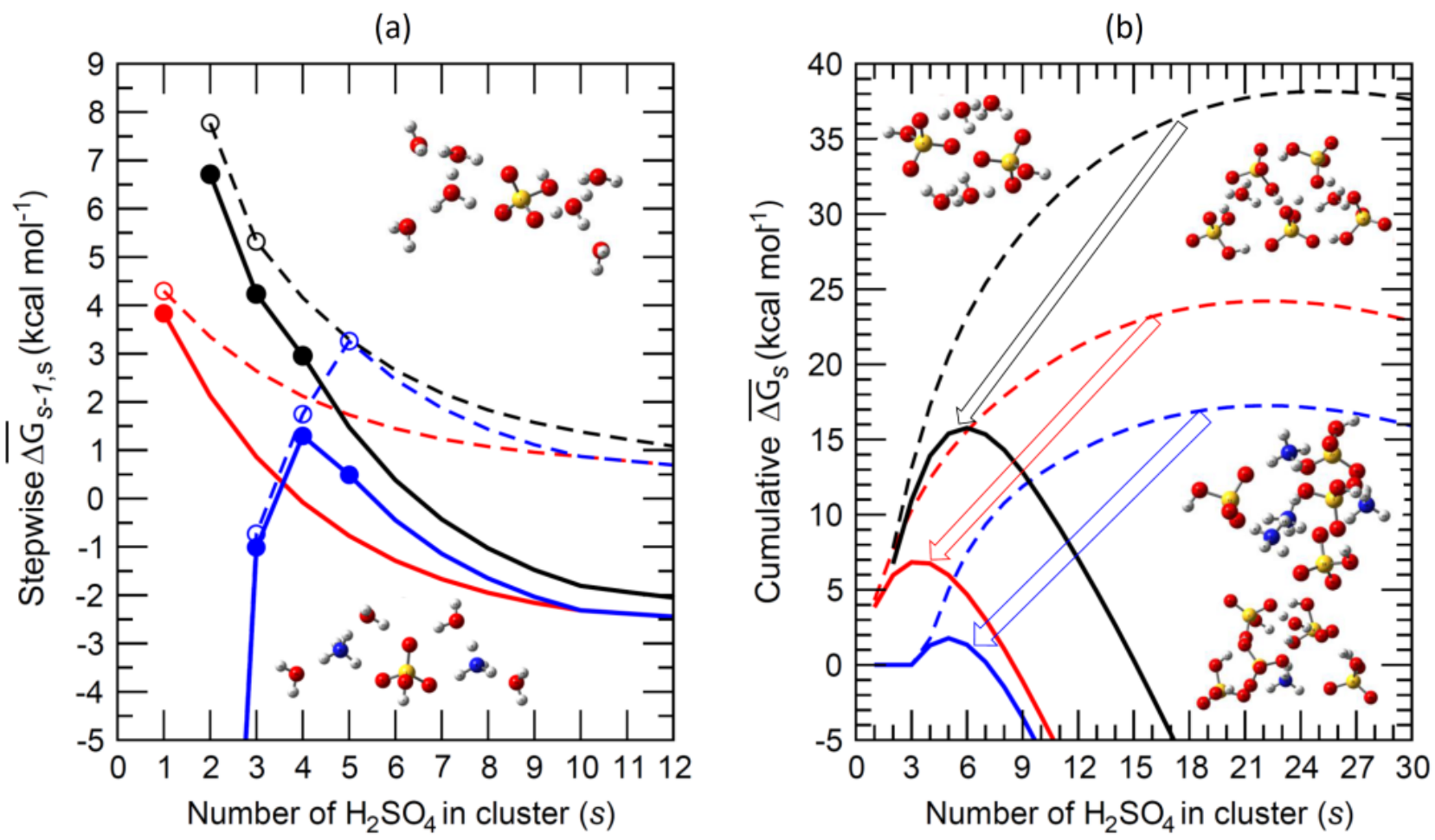

**Figure 4.** (a) Average stepwise Gibbs free energy change for the addition of one $H_2SO_4$ molecule to form a neutral (black), positively charged (red), or negatively charged (blue) binary $H_2SO_4$-$H_2O$ (dashed lines or empty circles) or ternary $H_2SO_4$-$H_2O$-$NH_3$ (solid lines or filled circles) cluster containing *s* $H_2SO_4$ molecules ($\overline{\Delta G}_{s-1,s}$); (b) Same as (a) but for the cumulative (total) Gibbs free energy change in each case. Filled and empty circles in (a) refer to $\overline{\Delta G}_{s-1,s}$ obtained using measurements and/or quantum-chemical calculations. $\overline{\Delta G}_{s-1,s}$ for larger clusters with $s \geq 10$, which approach the properties of the equivalent bulk liquid (20), are calculated using the capillarity approximation. Interpolation is used to calculate $\overline{\Delta G}_{s-1,s}$ for clusters up to $s=10$ (Eq. 11). Calculations were carried out at T=292 K, RH=38%, $[H_2SO_4]=3\times10^8$ cm$^{-3}$ and $[NH_3]$= 0.3 ppb. The inset diagrams represent equilibrium geometries for the most stable isomers of selected binary clusters ( $(H_3O^+)(H_2SO_4)(H_2O)_6$, $(H_2SO_4)_2(H_2O)_4$, and $(HSO_4^-)(H_2SO_4)_4(H_2O)_2$ ), and ternary clusters ( $(NH_4^+)(H_2SO_4)(NH_3)(H_2O)_4$, $(HSO_4^-)(H_2SO_4)_4(H_2O)(NH_3)$, $(H_2SO_4)_4(NH_3)_4$ ).

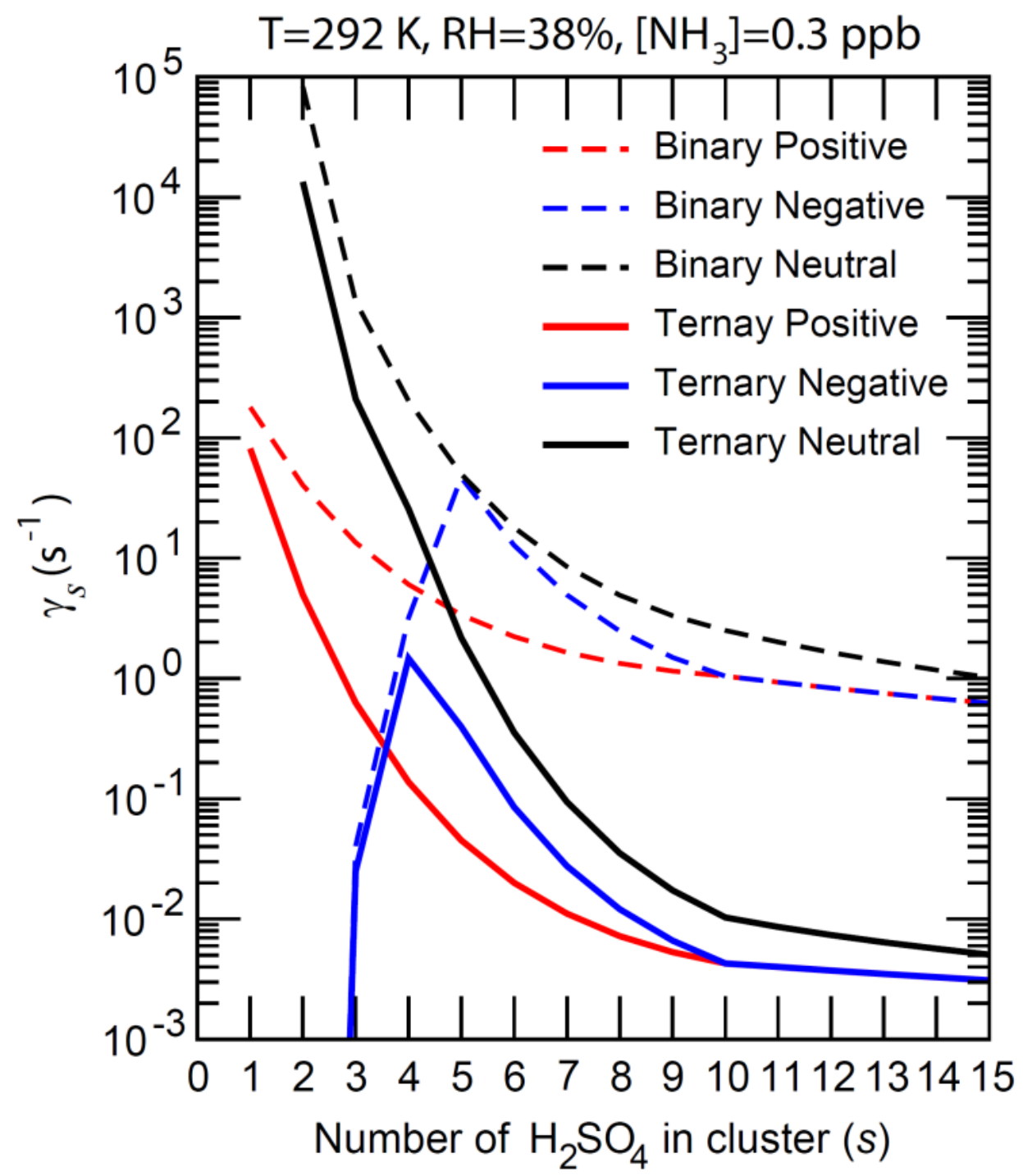

**Figure 5.** The number-concentration-weighted mean evaporation rates ($\bar{\gamma}$) of $H_2SO_4$ molecules from neutral clusters (black), positively charged clusters (red), and negatively charged clusters (blue) for binary ($H_2SO_4$-$H_2O$, dashed lines) and ternary ($H_2SO_4$-$H_2O$-$NH_3$, solid lines) nucleating systems containing $s$ $H_2SO_4$ molecules ($\overline{\Delta G}_{s-1,s}$). T=292 K, RH=38%, and [$NH_3$] = 0.3 ppb for the ternary system.

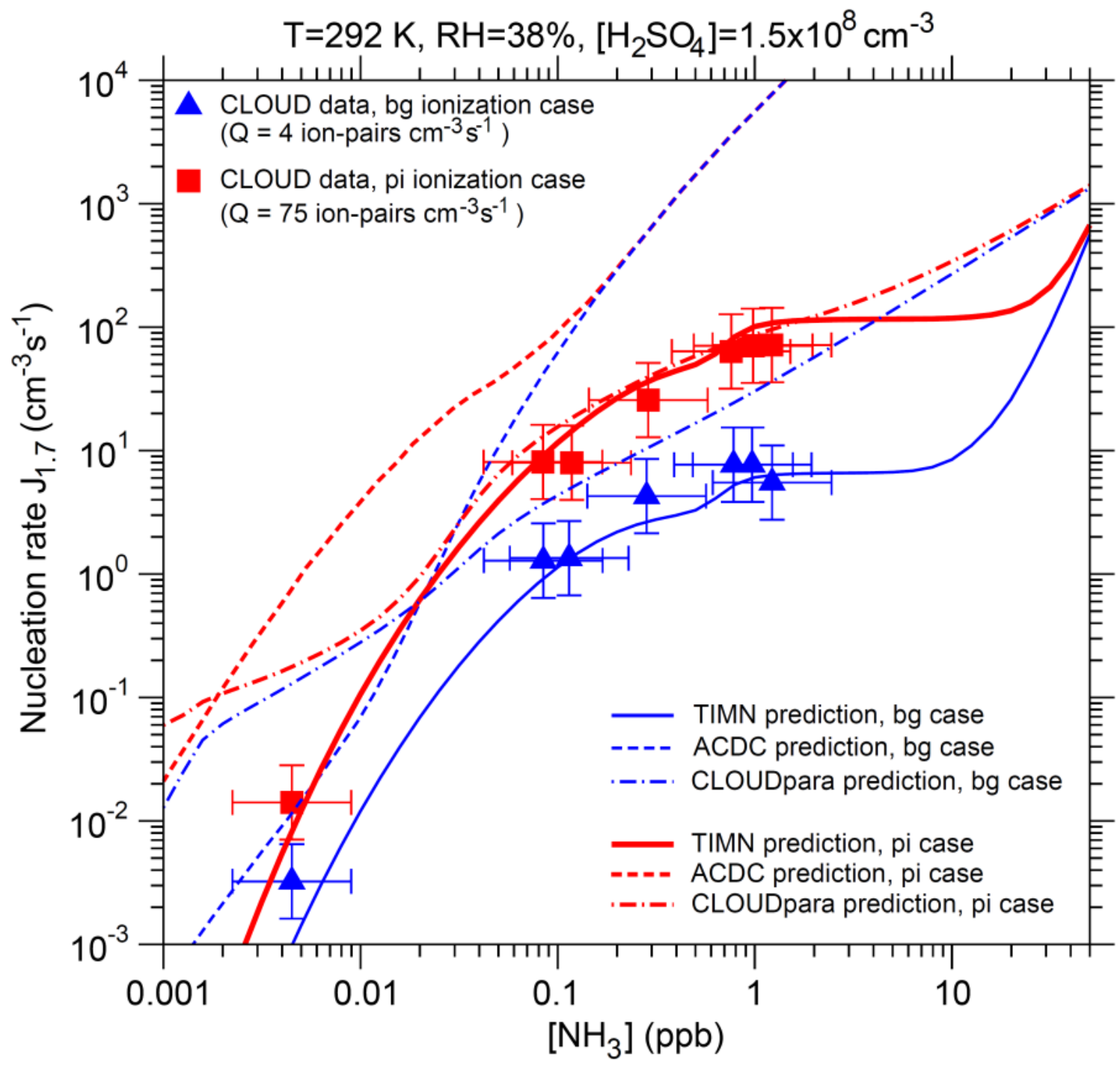

**Figure 6.** Effect of ammonia concentrations ([$NH_3$]) on effective nucleation rates calculated at a cluster mobility diameter of 1.7 nm ($J_{1.7}$, lines) under the stated conditions with two ionization rates (Q) – background ionization, bg (blue), and ionization enhanced by a pion beam, pi (red). Also shown are predictions from the TIMN model, the Atmospheric Cluster Dynamics Code (ACDC) (McGrath et al., 2012; Kurten et al., 2016), and an empirical parameterization of CLOUD measurements (CLOUDpara) (Dunne et al., 2016) are indicated by solid, dashed, and dot-dashed lines, respectively. The symbols refer to CLOUD experimental data (Kirkby et al., 2011; Dunne et al., 2016), with the uncertainties in measured [$NH_3$] and $J_{1.7}$ shown by horizontal and vertical bars, respectively. To be comparable, the CLOUD data points given in Dunne et al. (2016) under the conditions of T=292 K and RH=38% with [$H_2SO_4$] close to $1.5\times10^8$ $cm^{-3}$ have been interpolated to the same [$H_2SO_4$] value (=$1.5\times10^8$ $cm^{-3}$).

1021

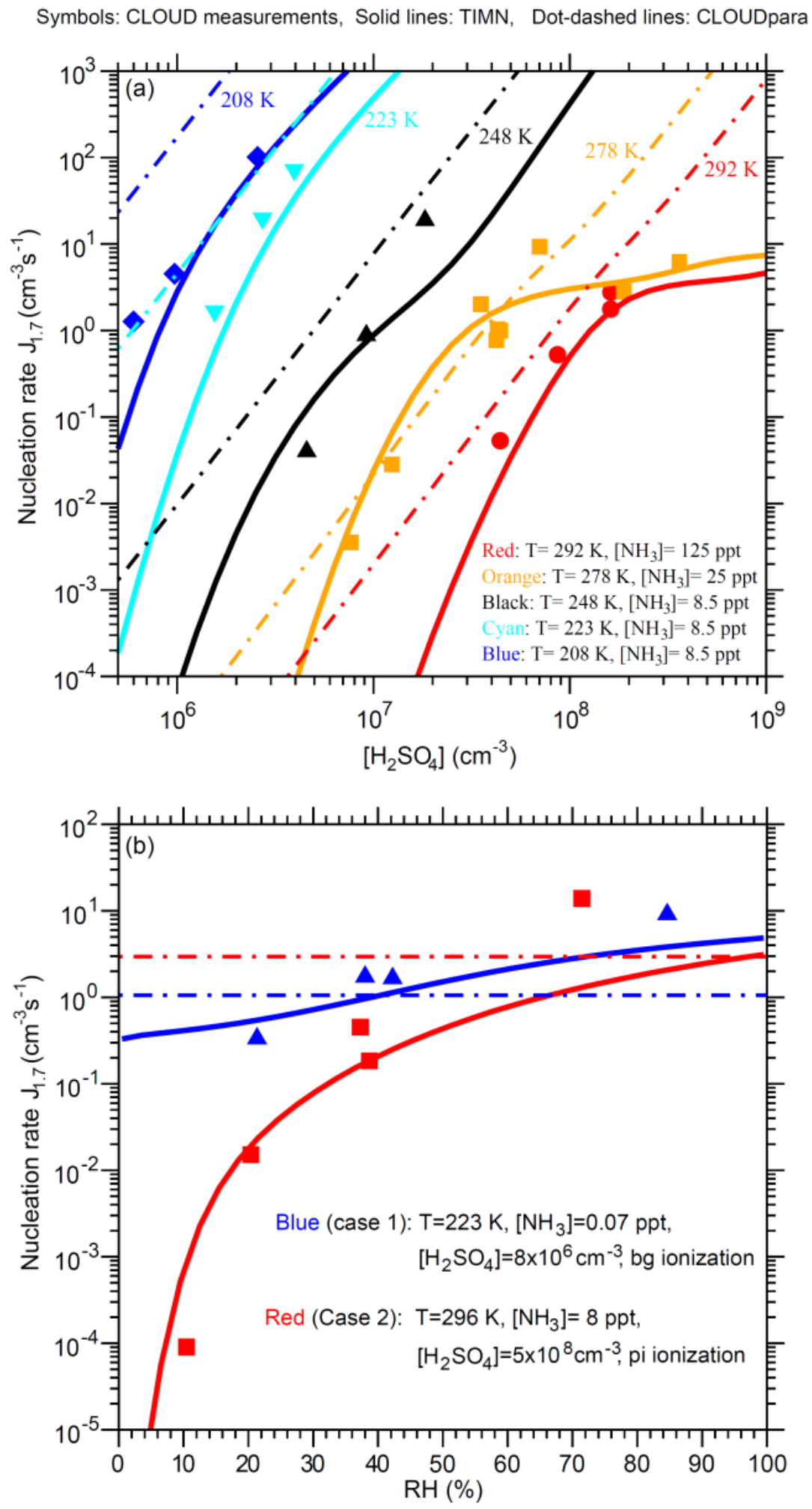

**Figure 7.** Comparison of TIMN simulations (solid lines), CLOUDpara predictions (Dunne et al.,
2016) (dot-dashed lines) and CLOUD measurements (symbols, data from Dunne et al. (2016) of
the dependences of nucleation rates on (a) $[H_2SO_4]$ at five different temperatures (T=292, 278,
248, 223, and 208 K) and (b) RH at two sets of conditions as specified. $[NH_3]$ is in ppt (parts per
trillion, by volume). Error bars for the uncertainties in measured $[H_2SO_4]$ (-50%, +100%), $[NH_3]$
(-50%, +100%), and $J_{1.7}$ (overall a factor of two) are not shown. To be comparable, the CLOUD
data points given in Dunne et al. (2016) under the conditions (T, RH, ionization rate) with $[NH_3]$
or $[H_2SO_4]$ close to the corresponding values specified in the figure legends have been interpolated
to the same $[NH_3]$ (Fig. 7a) or $[H_2SO_4]$ (Fig. 7b) values.
1031