# Peer review of "H2SO4-H2O-NH3 ternary ion-mediated nucleation (TIMN): Kinetic-based model and comparison with CLOUD measurements 3 Fangqun Yu1, Alexey B. Nadykto1, 2, Jason Herb1, Gan Luo1, Kirill M. Nazarenko2</sup"

_Atmospheric Chemistry and Physics, 2018_

## Referee Comment (RC1) · Anonymous Referee #1 · 20 Jun 2018

Review of "H$_2$SO$_4$-H$_2$O-NH$_3$ ternary ion-mediated nucleation (TIMN): Kinetic-based model and comparison with CLOUD measurements" by F. Yu et al.

The manuscript presents a kinetic, quasi-unary molecular cluster and aerosol particle model to simulate ternary ion-mediated nucleation (TIMN) from sulfuric acid (H$_2$SO$_4$), water (H$_2$O) and ammonia (NH$_3$). This work extends the previously developed binary H$_2$SO$_4$-H$_2$O (BIMN) model (Yu, 2006b) to include also ammonia. This is done by using quantum chemical data for some H$_2$SO$_4$-H$_2$O-NH$_3$ molecular clusters (some of which have been previously published by the authors, and some of which are new

but not yet published) and previously measured experimental thermodynamic data for bulk solutions, and implementing them in the model. Model results for the formation rate $J_{1.7}$ of nanoparticles of 1.7 nm are compared to rates derived from particle measurements at the CLOUD aerosol chamber.

The manuscript is fairly clearly written and suits in the scope of ACP. However, the model details need further clarification, and some assumptions and approximations made in the model require justification and/or more discussion. Also, discussion of the results with respect to experiments and other nucleation parameterizations or models needs to be more balanced. After the authors have addressed these issues (as listed in detail below), the study can be considered for publication in ACP.

Specific comments:

The most important issues regarding the model can be summarized as follows:

The authors claim to present "the first comprehensive kinetically-based $H_2SO_4$-$H_2O$-$NH_3$ ternary ion-mediated nucleation (TIMN) model that is based on the thermodynamic data derived from both quantum-chemical calculations and laboratory measurements."

However, it turns out that the model is in fact quasi-unary, i.e. approximates the multi-compound chemical system as a one-compound system. Also, the quantum-chemical data is rather sparse, liquid thermodynamic data is used from quite small nanoparticle sizes onward, and the rest of the thermodynamics is in practice guessed by connecting quantum chemical and bulk data by an exponential function.

These facts and the related uncertainties should be clearly brought up and discussed. Considering the roughness of some approximations, the suggestion that the model is in excellent agreement with CLOUD data needs much more comparisons and more than a few data points from CLOUD.
**Thermodynamic data**

The thermodynamic input data includes quantum chemical (QC) data for the very smallest clusters of a few molecules. Particles containing more than at least ten sulfuric acid molecules are assumed to behave according to the electrically neutral macroscopic liquid droplet model. For the intermediate sizes below ten $H_2SO_4$ molecules, QC data and liquid data are connected together by a type of exponential function.

(1) In general, it would be extremely helpful to explain the thermodynamic data in the form of a table which lists the different cluster / particle compositions and sizes, and the approaches used for their Gibbs free energies. It's much easier and faster for the reader than finding the information in the text.

*Quantum chemical (QC) data for small molecular clusters*

The QC data has been obtained using the PW91PW91/6-311++G(3df,3pd) density functional theory (DFT) method. PW91 is a commonly used DFT method in atmospheric cluster calculations, and it has been shown to yield mean errors similar to (although somewhat higher than) other common DFT methods in QC benchmarking studies (e.g. Elm and Kristensen, 2017).
In terms of the number of sulfuric acid molecules, which is the principal building block of the clusters and particles in the presented kinetic model, the used QC data covers cluster sizes up to (a) 1 sulfuric acid molecule for positively charged, (b) 4 for negatively charged (5 if the bisulfate ion is counted in), and (c) 4 for electrically neutral clusters.
(2) Page 5, line 152: "The thermodynamic data sets used for binary clusters were also updated."
For which clusters were the data updated: All or just some of them? What kind of differences are there compared to the previously published data for these clusters?

(3) Page 9, line 250: "We have extended the earlier QC studies of binary and ternary clusters to larger sizes."
Which sizes? Please indicate clearly which clusters are new, and which have been studied in previous publications. Also, list clearly the clusters for which QC data is applied (instead of other type of thermodynamic data). Are these the clusters listed in Tables 2-4?

(4) Page 9, lines 254-255: The authors have used also a "locally developed sampling code, which creates a 'mesh' around the cluster, in which molecules being attached to the cluster are the mesh nodes", but this sentence is all that is said about the code. Please elaborate what this code exactly does, and give a reference, if possible.

(5) Page 10, lines 289-292: "Both the absolute values and trends in $\Delta G^0_{+W}$ derived from calculations are in agreement with the laboratory measurements within the uncertainty range of $\sim$1-2 kcal mol$^{-1}$ for both QC calculations and measurements. This confirms the efficiency and precision of QC methods in calculating thermodynamic data needed for the development of nucleation models."
$\sim$1-2 kcal mol$^{-1}$ can be expected to be the general uncertainty of quantum chemical methods. However, as the Gibbs free energies are incorporated in the exponential factor of the evaporation rate (Eq. (7)), this uncertainty may propagate to an uncertainty of up to orders of magnitude in the particle formation rates and concentrations.
This is discussed and demonstrated e.g. by Kürten at al. (2016), who estimated the uncertainties in the modeled particle formation rates by increasing or decreasing all

Gibbs free energies by 1 kcal mol$^{-1}$. Depending on the conditions, this changes the formation rate by less than an order of magnitude, or even by up to several orders of magnitude.

Please discuss also the sensitivity of the particle kinetics to the evaporation rates, and the impact of the uncertainties in $\Delta G$ on the formation rate.

(6) Page 11, lines 321-322: "Since positively charged H$_2$SO$_4$ dimers are expected to contain large number of water molecules, no quantum chemical data for these clusters are available."

What does this mean? Does it mean that the data cannot be computed at all, or that the authors haven't computed such data in these studies?

(7) Page 12, lines 348-351: "Table 3 shows that the presence of NH$_3$ in H$_2$SO$_4$ clusters suppress hydration and that $\Delta G^0_{+W}$ for S$_2$A$_2$ falls below -2.0 kcal mol$^{-1}$. This is consistent with earlier studies by our group and others showing that large S$_n$A$_n$ clusters (n>2) are not hydrated under typical atmospheric conditions."

Please give references for these studies, especially for those conducted by other groups. Does this mean that all clusters and nanoparticles larger than (H$_2$SO$_4$)$_2$(NH$_3$)$_2$ are generally not hydrated, or do the particles become hydrated again at some larger size? At which size? What is assumed about the hydration of electrically neutral H$_2$SO$_4$-NH$_3$ clusters beyond the quantum chemistry data set, i.e. larger than (H$_2$SO$_4$)$_4$(NH$_3$)$_5$?

Generally, the hydration of a specific cluster (S$_2$A$_2$) tells nothing about the hydration of other clusters with different numbers of acid and base molecules. Therefore, it should be stated here that neglecting water in the larger clusters is just an assumption that has to be made due to the lack of thermodynamic data (as the authors have actually done later on line 454).

(8) Page 12, lines 365-367: "This finding is fully consistent with the laboratory measurements showing that growth of neutral $S_sA_a$ clusters follows $s = a$ pathway (Schobesberger et al., 2015)."
The study by Schobesberger et al. (2015) does not present any measurement data for neutral clusters. (Instead, they are modeled by the ACDC program in that study.) Please correct the sentence.

(9) Page 13, lines 389-393: "$\Delta G^0_{+S}$ values for $S^-S_{3-4}W_w$ are consistently $\sim$1.5-3 kcal mol$^{-1}$ less negative than the corresponding semi-experimental estimates (Table 4). The possible reasons behind the observed systematic difference are yet to be identified and include the use of low-level ab initio HF method to compute reaction enthalpies and uncertainties in experimental enthalpies in studies by Froyd and Lovejoy (2003)."
The computed values for $\Delta G^0_{+W}$ (as well as for $\Delta G^0_{+W}$ for some positive clusters), on the other hand, are *more* negative than those determined by Froyd and Lovejoy (2003). Why doesn't the discussed systematic difference apply to these values?

(10) Page 15, lines 471-472: "In the TIMN model, the equilibrium distributions are used to calculate number concentrations weighted stepwise Gibbs free energy change"
Where is this averaged $\Delta G$ used in the model? Doesn't the model use averaged evaporation frequencies (Eq. (10)), which are calculated over the individual evaporation rates and thus do not correspond to the averaged $\Delta G$?

(11) Page 15, lines 477-479: "In the atmosphere, where substantial nucleation is observed, the sizes of critical clusters are generally small ($s < \sim 5 - 10$) and nucleation rates are largely controlled by the stability (or $\gamma$) of small clusters with $s < \sim 5 - 10$."
Please give references for this.

(12) Table 2 and discussion on pages 17-19: For positively charged clusters, there is QC data only up to clusters containing one $H_2SO_4$ molecule and two $NH_3$ molecules. That is, not even the first growth step with respect to $H_2SO_4$ clustering (i.e. the formation of a $H_2SO_4$ dimer) is covered, and in practice all the data for positive clusters is guessed by using Eq. (11) (except for clusters containing more than 10 acid molecules, starting from which data for electrically neutral bulk solutions is used also for the positive clusters).
This is an extremely crude approximation. Please bring up this fact, and explain what new "insights" we can learn about the thermodynamics of these clusters by using these type of data.

(13) Page 17, lines 536-538: Similarly to positive clusters, the results for the thermody-namics of negative clusters raise some questions: "The effect of $NH_3$ on negative ions becomes important at $s \geq \sim 4$, when bonding between the clusters and $NH_3$ becomes strong enough to contaminate a large fraction of binary clusters with ammonia (Fig. 3)."
No QC data or experimental bulk data is used for the clusters around sizes $s \approx 4 - 9$. Considering that this behavior is deduced by interpolating between QC data for small clusters that take up ammonia rather weakly, and macroscopic solution data for an electrically neutral $H_2SO_4$-$H_2O$-$NH_3$ liquid, it is difficult to see this result as very reliable. Please state that the thermodynamics of these clusters are highly uncertain (or explain why they would not be).

(14) Table 4: The hydrate data for the negatively charged clusters is quite sparse for some clusters. Why are not hydrates with more water molecules considered for, for example, $S^-S_1$ or $S^-S_2$?

(15) Tables 2-4: Why isn't all QC data that is used in the model given in the tables?

For example, binary electrically neutral clusters are not included in Table 3. Please indicate clearly where these data can be found.

(16) Figure 3: Why are some hydrates with different numbers of water molecules grouped together? For instance, $(H_2SO_4)_2(H_2O)_{1-7}$ is presented as one bar; this doesn't tell much about the hydration as 1 and 7 are quite different numbers.
Also, in panel (d), please clarify that there is no hydrate data for these clusters; otherwise the figure panel might be understood so that the clusters don't take up water at all.

(17) Figure 4: Why is the cumulative Gibbs free energy zero for the first growth steps of the negative clusters in panel (b)? In panel (a), it does not look like these values add up to zero, but should be negative instead.

(18) Caption of Figure 4: "Calculations were carried out at T=292 K, RH=38%, $[H_2SO_4]=3x10^8$ cm$^{-3}$ and $[NH_3]$= 0.3 ppb."
How were the vapor concentrations, e.g. $[H_2SO_4]$, used in the calculations?

***Experimental bulk data for larger nanoparticles***

Bulk thermodynamic data is assumed for particles of all charging states containing at least ten $H_2SO_4$ molecules. While this is in practice the only available option due to the lack of other type of data, the approximation calls for some discussion about the related uncertainties.

(19) At which conditions (temperature, partial pressures of the $H_2SO_4$, $NH_3$, $H_2O$ vapors) were the measurements (Marti et al., 1997; Hyvärinen et al., 2005) performed?

How reliably can it be extrapolated to different conditions outside the measurement range?

(20) Page 16, lines 487-491: "Based on experimental data (Kebarle et al., 1967; Davidson et al., 1977; Wlodek et al., 1980; Holland and Castleman, 1982; Froyd and Lovejoy, 2003), the stepwise $\Delta G$ values for clusters decreases exponentially as the cluster sizes increase and approaches to the bulk values when clusters containing more than ~8-10 molecules (Yu, 2005)."
Is possible size-dependent chemical composition, i.e. acid:base molar ratio, considered here (e.g. Chen et al., 2018)? How does it affect the model results?

(21) Page 16, lines 491-494: "Cluster compositions measured with an atmospheric pressure interface time-of-flight (APi-TOF) mass spectrometer during CLOUD experiments also show that the chemical effect of charge-carrying becomes unimportant when the cluster contains more than 9 $H_2SO_4$ molecules (Schobesberger et al., 2015)."
In the study by Schobesberger et al. (2015), it looks like the different charges approach similar composition somewhere in the size range where the $H_2SO_4$ content is ~20-100 molecules (Figure 9 in the study). At 10 $H_2SO_4$ molecules, the composition of negative and positive particles is still different. Please comment.

(22) Page 17, line 524: "consistent with the laboratory measurements (Marti et al., 1997)"
Isn't the discussed $\Delta G$ data derived from these measurements (i.e. naturally, it is consistent)? Please clarify.

Page 24, lines 774-776: Is this the correct reference?

**Approximated values for intermediate sizes with < 10 H$_2$SO$_4$ molecules**

(23) Eq. (11): What is this "extrapolation" formula based on? It is not clear why this functional form would be suitable for connecting QC and bulk measurements. Please explain clearly how the formula is derived, and discuss the related uncertainties.

(24) Page 16, lines 503-506: "$c$ in Eq. (11) is the exponential coefficient that determines how fast $\Delta G_{s-1,s}$ approaches to bulk values as $s$ increases. In the present study, $c$ is estimated from $\Delta G_{s-1,s}$ at $s$=2 and $s$=3 for neutral binary and ternary cluster for which experimental (Hanson and Lovejoy, 2006; Kazil et al., 2007) or quantum-chemical data (Table 3) are available."
What can the data for clusters that contain 2 or 3 H$_2$SO$_4$ molecules possibly tell about how fast $\Delta G$ approaches bulk values?
Is $c$ estimated based on QC data, experimental data, or both? How is this done exactly? Is it only for neutral clusters, or also for charged clusters?

(25) Finally, the most important issue regarding the thermodynamics is the fact that the "critical sizes", i.e. the barriers for nucleation, are located around cluster sizes for which there is no reliable thermodynamic data (Figure 4). For all different types of clusters (binary, ternary, all charging states), the maximum of the free energy curve is beyond the QC data (or just at the upper limit of the QC data in the case of negative ternary clusters). That is, the critical stage of nucleation is based on Eq. (11), which in turn does not seem to be based on an actual physical model.
Considering this, can the model really give important new information on H$_2$SO$_4$-H$_2$O-NH$_3$ particle formation mechanisms?
**Kinetic model**

(26) In the kinetic model, the clusters are assumed to be in equilibrium with respect to both water and ammonia. Such equilibriation assumption can be made if the time scales of the attachment and evaporation processes of some compound are substantially shorter than those of other compounds. This is the case for water, as (a) its concentration (to which the attachment, i.e. molecular collision, frequency is directly proportional) is around $\sim$10 orders of magnitude higher than that of $H_2SO_4$ or $NH_3$, and (b) its binding to the clusters is so much weaker that its evaporation rate is $\sim$ several orders of magnitude higher than that of other compounds (except for some charged clusters in e.g. Table 2).

This is, however, generally not the case for ammonia. The binding of $NH_3$ depends strongly on the cluster composition: Depending on the acid:base ratio, either $NH_3$ or $H_2SO_4$ evaporates much faster than the other. Within the set of small clusters, the weakest and strongest bindings of $NH_3$ are of the same order as those of $H_2SO_4$ (e.g. Table 3). The collision rates of $NH_3$ are not necessarily multiple times higher than those of $H_2SO_4$, either: While ammonia is generally more abundant than $H_2SO_4$ in the atmosphere, there are environments where $[H_2SO_4]$ and $[NH_3]$ are around the same order (such as some of those simulated in this study).

Due to these reasons, the explanation for assuming equilibrium with respect to $NH_3$ is not justifiable (pages 13-14, lines 414-418): "In the lower troposphere, where most of the nucleation events were observed, $[H_2SO_4]$ is typically at sub-ppt to ppt level, while $[NH_3]$ is in the range of sub-ppb to ppb levels. This means that small ternary clusters can be considered to be in equilibrium with $H_2O$ and $NH_3$ vapors."

(a) Doesn't ammonia need to also evaporate much faster from the clusters for the equilibrium assumption to be justified? (b) At the simulated conditions, $[H_2SO_4]$ and $[NH_3]$ are in many cases of the same order. For instance, in Figure 6 at the lower end of the $[NH_3]$ axis, $[NH_3]$ is of the same order or even lower than $[H_2SO_4]$. In Figure 7a, $[NH_3]$ is around 10 ppt, i.e. $\sim 10^8$ cm$-3$, and $[H_2SO_4]$ is around $10^7 \ldots 10^9$ cm$^{-3}$.

**Please show that the equilibrium with respect to NH$_3$ really is a valid assumption for these simulation systems.**

(27) Some other aspects of the model also need clarification. The kinetic equations (Eqs.(1-6)) seem to include also collisions between charged clusters / particles of the same polarity. How high are the rate constants for such processes? Doesn't electrostatic repulsion prevent these attachments?

Further, if multiply charged particles can form in these collisions, how are these different charge numbers treated in the model? Shouldn't there be separate equations for particles that contain a single charge, two charges, three, and so on?

(28) Page 5, lines 162-166: "The initial negative ions, which are normally assumed to be NO$_3^-$, are converted into HSO$_4^-$ core ions (i.e., S$^-$) and, then, to larger H$_2$SO$_4$ clusters in the presence of gaseous H$_2$SO$_4$. The initial positive ions H$^+$W$_w$ are converted into H$^+$A$_{1-2}$W$_w$ in the presence of NH$_3$, H$^+$S$_s$W$_w$ in the presence of H$_2$SO$_4$, or H$^+$A$_a$S$_s$W$_w$ in the case, when both NH$_3$ and H$_2$SO$_4$ are present in the nucleating vapors."

What are the rate constants for the conversions of NO$_3^-$ and H$^+$W$_w$? What does H$^+$A$_{1-2}$W$_w$ (or H$^+$S$_s$W$_w$ and H$^+$A$_a$S$_s$W$_w$) mean, i.e. how many ammonia and water molecules does it contain?

In the equations (page 7, lines 192-193), "$N_0^{+,-}$ and $Q$ are the concentration of initial ions not containing H$_2$SO$_4$ and the ionization rate, respectively"

What do the "initial ions" refer to, e.g. H$^+$W$_w$ or H$^+$A$_{1-2}$W$_w$? NO$_3^-$ or HSO$_4^-$?

(29) Eq. (3): Why does the evaporation term for creating H$_2$SO$_4$ monomers from a dimer includes a factor of two ($\delta_{j,2}$), but the corresponding collision term, removing monomers in the collision creating a dimer, does not?

(30) Page 8, lines 212-213: "The methods for calculating $\beta$, $\gamma$, $\eta$, and $\alpha$ for binary $H_2SO_4$-$H_2O$ clusters have been described in detail in Yu (2006b)."
I was not able to find the descriptions for $\beta$, $\eta$, and $\alpha$ in the given reference; the paper only seems to re-direct the reader to discussion in 3 other papers. Please briefly summarize how these parameters are obtained.

(31) Page 8, lines 221-222: "$N^o$ is the number concentration of $H_2SO_4$ at a given $T$ under the reference vapor pressure $P$ of 1 atm."
Isn't $N^o$ simply the number concentration corresponding to the reference pressure $P$ of the QC calculations? What does it have to do with any [$H_2SO_4$]? In general, the evaporation rate should not be related to the concentration of any compound, as it does not depend on the composition of surrounding vapor (only on the temperature, i.e. the inert gas).

(32) Page 8, lines 223-225: "The temperature dependence of $\Delta H^o$ and $\Delta S^o$, which is generally small and typically negligible over the temperature range of interest, was not considered."
Can you give a reference for the negligible temperature dependence?

(33) Page 19, lines 572-573: "mean evaporation rate ($\bar{\gamma}$) of an $H_2SO_4$ molecule"
Is it assumed that only a single $H_2SO_4$ molecule evaporates, i.e. no water ligands, for instance, are attached to it? If so, please discuss the validity of this assumption, or even better, average the evaporation rates over all evaporation pathways with different numbers of other compounds attached to the acid molecule.

(34) Page 19, lines 573-574: "The shapes of $\bar{\gamma}$ curves are similar to those of $\Delta\bar{G}_{s-1,s}$ (Fig. 4a) as $\bar{\gamma}$ values are largely controlled by $\Delta\bar{G}_{s-1,s}$."

How is $\bar{\gamma}$ related to the averaged values $\bar{\Delta G}_{s-1,s}$? Isn't $\bar{\gamma}$ calculated based on individual values $\Delta G_{s-1,s}$ (Eq. (10)), i.e. not exactly equivalent to $\bar{\Delta G}_{s-1,s}$?

(35) The discussion on page 19, lines 575-584, feels somewhat confusing: First it's said that the effect of ammonia is significant for larger clusters and of less importance for small clusters (e.g. "the binding of $NH_3$ to small neutral and charged clusters are weaker compared to that for larger clusters"), but after this it's concluded that "The nucleation rates, limited by formation of small clusters ($s <\sim 5$), depend strongly on the stability or evaporation rate of these small clusters and, thus, on $[NH_3]$."
So is or is not $NH_3$ important for the small clusters and nucleation? Please clarify.

(36) Page 19, line 588: "the concentrations of clusters of all sizes are explicitly predicted"
A quasi-unary model cannot be called "explicit"; please re-formulate.

(37) Eq. (13): Is it so that only growth through $H_2SO_4$ vapor is taken into account in the calculation of the particle formation rate? What about the effects of coagulation and recombination?
The quantity $J$ that can be deduced from measurements -and that also is the relevant quantity for atmospheric modeling- includes all processes through which particles form, not only monomer condensation and evaporation. Therefore, these should be included also in the model-based formation rate.

(38) Figure 1: The figure is confusing, and using patterns to fill the lines or spheres makes it somewhat difficult to read. For instance, it looks like "Condensation" means that electrically neutral clusters are ionized into charged particles (the arrows lead only to the charged blocks), and that "Coagulation / Scavenging" means that positively

charged particles attached to each other or neutral particles. What is the difference between "Coagulation / Scavenging" and "Coagulation"?

**Results and discussion**

(39) As a general comment, the description of the model should be a bit less ambitious. As one-compound discrete-sectional kinetic models have existed at least since the 1970s, the model cannot be considered "first", nor is it exactly "comprehensive" or "accurate" due to the quasi-unary assumption.

The addition of $NH_3$ to the previous BIMN model does not make the model very new, either, as it means simply using different thermodynamic data in an existing model - and the main author has also previously published a modeling study entitled "Effect of ammonia on new particle formation: A kinetic $H_2SO_4$-$H_2O$-$NH_3$ nucleation model constrained by laboratory measurements" (Yu, 2006a). Besides, as the authors themselves also bring up, the kinetics of $H_2SO_4$-$H_2O$-$NH_3$ molecular clusters including the different charging states have been previously modeled e.g. by the ACDC program (which the authors quite extensively criticize).

(40) As previously (e.g. Nadykto et al., 2011; Nadykto et al., 2014), the main criticism is targeted at the modeling work by University of Helsinki (and this time also at the particle formation rate parameterization CLOUDpara based on the experimental data from the CLOUD chamber). In general, the authors criticize the ACDC model; however, the output of a clustering model is determined by the input parameters, namely the thermodynamic data. The ACDC program does not use any specific QC data, but the data is instead given by the user.

The ACDC data presented by Kürten et al. (2016) results from QC thermochemistry calculated with the RI-CC2/aug-cc-pV(T+d)Z//B3LYP/CBSB7 method. Therefore, the authors should call this rather e.g. "RI-CC2//B3LYP" data than "ACDC" data. The

RI-CC2//B3LYP method is known to have a tendency to over-predict cluster stability, as has been discussed for example by the Helsinki group (e.g. Kupiainen-Määttä et al., 2015; Myllys et al., 2016), and thus it is not much used anymore in QC calculations.

(41) Page 4, lines 122-123: "ACDC is also an acid–base reaction model, with the largest clusters containing 4-5 acid and 4-5 base molecules (no water molecules)":
This is not the case, as ACDC is simply a program that solves the kinetic equations (similar to Eqs. (1-6)) for a given set of molecular clusters using given thermodynamic input data, which does not need to involve acids or bases. It is not limited to some fixed specific largest cluster sizes; in the cited studies, the largest sizes were determined by the availability of QC data for the systems of interest.

(42) Page 4, lines 127-130: "In ACDC, the nucleation rate is calculated as the rate of clusters growing larger than the upper bounds of the simulated system (i.e., clusters containing 4 or 5 $H_2SO_4$ molecules) (Kurten et al., 2016) and thus may over-predict nucleation rates when critical clusters contain more than 5 $H_2SO_4$ molecules."
It is of course not reasonable to model a system where the critical size region is outside the system boundaries. Thus, this region should be examined before simulating given conditions, as also discussed in the study by Olenius et al. (2013).

(43) Page 4, lines 130-132: "All clusters simulated by the ACDC model do not contain $H_2O$ molecules and the effect of relative humidity (RH) on nucleation thermochemistry is neglected." Page 21, lines 645-646: "an important influence of RH on nucleation rates (which is neglected in both the CLOUDpara and ACDC models)"
The authors of the present manuscript are well aware of the fact that water can be included in the ACDC model: in fact, the effect of cluster hydration was recently the topic of a rather heated discussion between these authors and the researchers at University of Helsinki (Nadykto et al., 2014; Kupiainen-Määttä et al., 2015; Nadykto

et al., 2015; in this case, the question was about $H_2SO_4$-dimethylamine clusters), including i.a. ACDC simulations conducted as a function of RH.

Hydration can naturally be included in a kinetic model, such as ACDC, given that there is thermodynamic input data for clusters containing water. Please correct your claims about this. The effect of water in the $H_2SO_4$-$H_2O$-$NH_3$ system has been studied by ACDC e.g. by Henschel et al. (2016).

(44) Also the particle formation rate parameterization by Dunne et al. (2016) is criticized. It would be fair to note that the deviations of the parameterization from the CLOUD data are not a new finding, as the uncertainties and weaknesses of the parameterization are discussed rather extensively in the work by Dunne et al. (e.g. supplementary Figures S3-S6).

(45) Page 11, line 333-334: "most of these studies, except for Nadykto and Yu (2007), did not consider the impact of $H_2O$ on cluster thermodynamics"

The effect of $H_2O$ on $H_2SO_4$-$NH_3$ clusters containing up to three $H_2SO_4$ and three $NH_3$ molecules has been considered by Henschel et al. (2014; 2016).

(46) Page 13, lines 396-397: The sentence "The binding of the second $NH_3$ to $S^-S_3A$ to form $S^-S_3A_2$ is much weaker than that of the first $NH_3$ molecule. This indicates that most of $S^-A_a$ can only contain one $NH_3$ molecule" isn't clear: How does the binding of $NH_3$ to a cluster containing 3 $H_2SO_4$ molecules indicate something about the attachment of $NH_3$ to a bisulfate ion $S^-$?

(47) Comparisons to CLOUD data (Figures 6 and 7): Many of the comparisons look quite nice indeed. However, **more experimental data over a wider range of conditions should be shown to support the claim that the model is "in excellent**

**agreement with CLOUD measurements"**.

For instance, in the work by Kürten et al. (2016) on CLOUD-based $J_{1.7}$, the model used in the study (ACDC with input thermodynamics computed with the RICC2//B3LYP method) is at some conditions in excellent agreement with CLOUD data, and at some conditions there are significant differences.

Therefore, comparisons with CLOUD data should be shown for **a large set of data**, for example the figures of the study by Kürten et al. (2016), including also electrically neutral cases and a wider range of ammonia concentrations.

(48) Figure 6: The original CLOUD data includes also $J_{1.7}$ for experiments with no ions. Please add these electrically neutral experimental and model data to the figure. It looks like the slope of the modeled $J_{1.7}$ is quite steep when neutral nucleation takes over; it is interesting to see how this compares with the measurements.

(49) Figure 7, top panel: For most lines, there are only 3 experimental data points, which doesn't make the comparison of these data to the model lines very strong. As there is so much CLOUD data available, please pick more representative data from e.g. the work by Kürten et al. (2016).

Especially low but still non-negligible ammonia mixing ratios are not shown in the current comparisons. If the model is said to cover "a wide range of atmospheric conditions", these should be included.

Technical comments:

(50) Change all occurrences of "physio-chemical" to "physico-chemical"; presumably "physio" refers to physiology, not physics.

(51) Page 2, line 35: Change "specie" to "species".

(52) Page 9, lines 240-245: The sentence "In earlier studies, this method has been applied to a large variety of atmospherically-relevant clusters and has been shown to be well suited to study the ones, (...)" is clumsy (i.e. what does "the ones" refer to?); please re-formulate.

(53) Page 9, line 253: Change "basin hoping" to "basin hopping".

(54) Page 11, line 332: It is misleading to list Kürten et al. (2015) as a computational study, as it doesn't present any computationally obtained thermodynamics.

(55) Page 16, line 505: Change "cluster" to "clusters".

(56) Table 1: Please give units for the energy quantities. Please also clarify that "H" and "S" may refer to either the energetics, or the cluster composition (the first column), or use different symbols for some of the abbreviations / quantities. Also change "based" in the footnote to "based on".

(57) The resolution and/or clarity of some figures, mainly 1 and 3, is rather poor. Please fix this.

**References (only those not included in the reference list of the manuscript):**

Chen et al., Aerosol Science and Technology, doi:10.1080/02786826.2018.1490005 (2018)

Elm and Kristensen, Physical Chemistry Chemical Physics 19, 1122-1133, doi:10.1039/c6cp06851k (2017)

Henschel et al., Journal of Physical Chemistry A 118, 2599-2611, dx.doi.org/10.1021/jp500712y (2014)

Henschel et al., Journal of Physical Chemistry A 120, 1886-1896, doi:10.1021/acs.jpca.5b11366 (2016)

Kupiainen-Määttä et al., Chemical Physics Letters 624, 107-110, https://doi.org/10.1016/j.cplett.2015.01.029 (2015)

Nadykto et al., Entropy 13, 554-569, doi:10.3390/e13020554 (2011)

Nadykto et al., Chemical Physics Letters 609, 42-49, https://doi.org/10.1016/j.cplett.2014.03.036 (2014)

Nadykto et al., Chemical Physics Letters 624, 111-118, https://doi.org/10.1016/j.cplett.2015.01.028 (2015)

Myllys et al., Journal of Physical Chemistry A 120, 621-630, doi:10.1021/acs.jpca.5b09762 (2016)

---

## Referee Comment (RC2) · Anonymous Referee #3 · 30 Aug 2018

The kind of model introduced in this paper is definitely needed in atmospheric new particle formation research, so I am in principle in favor of publishing this work. I have, however, a few concerns that should be addressed before accepting the paper for publication.

I am not fully comfortable with the current structure of the paper. Sections 1 and 2.1 provide a nice introduction and background for this work. Section 2.2 is a compact description of the model and fine as well. Section 2.3 is, however, a mixture of technical details, model evaluations and scientific results/findings. I would prefer separating these issues to the extend possible. For example, the technical details related to the

used thermodynamic and other data as well as QC calculations could be put into a separate Appendix/Appendicies. Such details are a very important part of this paper, but not of major interest to most of the readers.

The authors state that a detailed description of QC calculations will be reported in separate papers. The authors should be very careful in this regard: this paper needs to have enough material to justify the obtained results.

Minor issues:

Please add to the text (line 197) that PH2SO4 refers to gas-phase production of sulfuric acid (in the atmosphere, sulfuric acid/sulfate can also be produced in liquid/aerosol phase).

The given ammonia concentration levels (beginning of section 2.4.1) should be backed up with suitable references. The authors should better justify the statement that small ternary clusters can be considered to be in equilibrium with ammonia. Mentioning solely the typical ammonia concentrations is not enough.

CLOUD should be defined also in the abstract.

There are a small number of grammatical issues that should be corrected, e.g. indicating (line 64), a nucleation model (line 67), did not (line 107), a similar pattern (line 296), the s=a pathway (line 367), even when they (line 384), under the condition??? (line 559).
* * *

---

## Author Comment (AC1) · 25 Oct 2018

Please see attached file.

The authors would like to thank the reviewer for the constructive comments. Our replies to the comments are given below, with the original comments in black, and our response in blue. We have revised the manuscript accordingly. All changes made to the manuscript have been marked with Track-Change tool in one of submitted files.

**Anonymous Referee #3**

The kind of model introduced in this paper is definitely needed in atmospheric new particle formation research, so I am in principle in favor of publishing this work. I have, however, a few concerns that should be addressed before accepting the paper for publication.

I am not fully comfortable with the current structure of the paper. Sections 1 and 2.1 provide a nice introduction and background for this work. Section 2.2 is a compact description of the model and fine as well. Section 2.3 is, however, a mixture of technical details, model evaluations and scientific results/findings. I would prefer separating these issues to the extend possible. For example, the technical details related to the used thermodynamic and other data as well as QC calculations could be put into a separate Appendix/Appendicies. Such details are a very important part of this paper, but not of major interest to most of the readers.

This is a good point. Following the referee's suggestion, we have moved some of the technical details related to the used thermodynamic and other data as well as QC calculations to supplementary material.

The authors state that a detailed description of QC calculations will be reported in separate papers. The authors should be very careful in this regard: this paper needs to have enough material to justify the obtained results.

This paper contains the adequate materials (as provided in the tables now in the supplementary material) to justify the obtained results. To address the reviewer's concern, we have deleted this sentence.

Minor issues:

Please add to the text (line 197) that PH2SO4 refers to gas-phase production of sulfuric acid (in the atmosphere, sulfuric acid/sulfate can also be produced in liquid/aerosol phase).

Modified as suggested.

The given ammonia concentration levels (beginning of section 2.4.1) should be backed up with suitable references. The authors should better justify the statement that small ternary clusters can be considered to be in equilibrium with ammonia. Mentioning solely the typical ammonia concentrations is not enough.

We have added several references about the ammonia concentration levels. We have also added discussions about the validity of the equilibrium assumption.

CLOUD should be defined also in the abstract.

Done.

There are a small number of grammatical issues that should be corrected, e.g. indicating (line 64), a nucleation model (line 67), did not (line 107), a similar pattern (line 296), the s=a pathway (line 367), even when they (line 384), under the condition??? (line 559).

Thanks for the careful reading. We have fixed these grammatical issues.

**Fig. 1.**

---

## Author Comment (AC2) · 26 Oct 2018

Thanks to the referee for very helpful comments, which have allowed us to clarify and improve the manuscript. Below we address the reviewer' comments, with the original comments in black, and our response in blue. We have revised the manuscript accordingly. All changes made to the manuscript have been marked with Track-Change tool in one of submitted files.

**Anonymous Referee #1**
Review of "$H_2SO_4$-$H_2O$-$NH_3$ ternary ion-mediated nucleation (TIMN): Kinetic-based model and comparison with CLOUD measurements" by F. Yu et al.
The manuscript presents a kinetic, quasi-unary molecular cluster and aerosol particle model to simulate ternary ion-mediated nucleation (TIMN) from sulfuric acid ($H_2SO_4$), water ($H_2O$) and ammonia ($NH_3$). This work extends the previously developed binary $H_2SO_4$-$H_2O$ (BIMN) model (Yu, 2006b) to include also ammonia. This is done by using quantum chemical data for some $H_2SO_4$-$H_2O$-$NH_3$ molecular clusters (some of which have been previously published by the authors, and some of which are new but not yet published) and previously measured experimental thermodynamic data for bulk solutions, and implementing them in the model. Model results for the formation rate $J_{1.7}$ of nanoparticles of 1.7 nm are compared to rates derived from particle measurements at the CLOUD aerosol chamber.
The manuscript is fairly clearly written and suits in the scope of ACP. However, the model details need further clarification, and some assumptions and approximations made in the model require justification and/or more discussion. Also, discussion of the results with respect to experiments and other nucleation parameterizations or models needs to be more balanced. After the authors have addressed these issues (as listed in detail below), the study can be considered for publication in ACP.

We appreciate the time and effort of the referee in providing the detailed comments. Please see below for our point-to-point replies and clarifications.

Specific comments:
The most important issues regarding the model can be summarized as follows:
The authors claim to present "the first comprehensive kinetically-based $H_2SO_4$-$H_2O$-$NH_3$ ternary ion-mediated nucleation (TIMN) model that is based on the thermodynamic data derived from both quantum-chemical calculations and laboratory measurements."
However, it turns out that the model is in fact quasi-unary, i.e. approximates the multicompound chemical system as a one-compound system. Also, the quantum-chemical data is rather sparse, liquid thermodynamic data is used from quite small nanoparticle sizes onward, and the rest of the thermodynamics is in practice guessed by connecting quantum chemical and bulk data by an exponential function.
These facts and the related uncertainties should be clearly brought up and discussed. Considering the roughness of some approximations, the suggestion that the model is in excellent agreement with CLOUD data needs much more comparisons and more than a few data points from CLOUD.

We feel that the referee probably misunderstood the TIMN model. As shown in Figures 1 and 3 and discussed in the text, the model is multicomponent and does not approximate multicomponent systems by one-component system. First of all, the distributions of small clusters of variable chemical composition were explicitly calculated (Figure 3) as a function of T, RH, and [$NH_3$]. Secondly, the compositions of neutral, positively charged and negatively charged clusters studied here are different. Thirdly, the model explicitly accounts for the formation and properties of both binary and ternary clusters and the interactions between neutral and charged clusters.

As for the amount of quantum-chemical data used to constrain the model, the TIMN model is constrained by largest amount of QC data available at the present time that was obtained using PW91PW91/6-311++G(3df,3pd) method. We have pointed out clearly in the manuscript that since the formation of small clusters is the limiting step for nucleation, improving nucleation thermodynamics by applying QC data is critically important. While extrapolation may lead to possible uncertainties which has been clearly acknowledged in the original manuscript, this approach provides nucleation thermochemistry of much better quality than conventional bulk liquid/capillarity approximation, which fails to predict free energies and formation rates of small molecular clusters, and is innovative in terms of connecting thermochemical properties of QC data for small binary and ternary clusters that cannot be adequately described by the capillarity approximation with those for large clusters that can be adequately described the very same capillarity approximation. In order to address the Reviewer's concern, additional discussion on uncertainties associated with the interpolation has been added to the revised manuscript.

As for the comparison with CLOUD data, Figures 6 & 7 show clearly that we have compared our model predictions with 48 data points from CLOUD measurement in the original manuscript. The comparisons include the dependences of nucleation rates on all the key parameters controlling nucleation rates: [$NH_3$], ion production rate, [$H_2SO_4$], temperature, and RH. In order to address the Reviewer's concern, we have made additional comparisons with CLOUD data and included them in the revised manuscript.

**Thermodynamic data**
The thermodynamic input data includes quantum chemical (QC) data for the very smallest clusters of a few molecules. Particles containing more than at least ten sulfuric acid molecules are assumed to behave according to the electrically neutral macroscopic liquid droplet model. For the intermediate sizes below ten $H_2SO_4$ molecules, QC data and liquid data are connected together by a type of exponential function.
(1) In general, it would be extremely helpful to explain the thermodynamic data in the form of a table which lists the different cluster / particle compositions and sizes, and the approaches used for their Gibbs free energies. It's much easier and faster for the reader than finding the information in the text.

Agreed. QC data are already in the form of a table (see Tables 2-4).

**Quantum chemical (QC) data for small molecular clusters**
The QC data has been obtained using the PW91PW91/6-311++G(3df,3pd) density functional theory (DFT) method. PW91 is a commonly used DFT method in atmospheric cluster calculations, and it has been shown to yield mean errors similar to (although somewhat higher than) other common DFT methods in QC benchmarking studies (e.g. Elm and Kristensen, 2017).
In terms of the number of sulfuric acid molecules, which is the principal building block of the clusters and particles in the presented kinetic model, the used QC data covers cluster sizes up to (a) 1 sulfuric acid molecule for positively charged, (b) 4 for negatively charged (5 if the bisulfate ion is counted in), and (c) 4 for electrically neutral clusters.
(2) Page 5, line 152: "The thermodynamic data sets used for binary clusters were also updated." For which clusters were the data updated: All or just some of them? What kind of differences are there compared to the previously published data for these clusters?

We have meant that the scheme to calculate the evaporation rates of binary clusters has been updated. In the previous IMN model (Yu, ACP, 2006), the evaporation rates of binary clusters were calculated with an equation considering the Thomson effect and dipole-charge interaction

(Eq. 14 in Yu (2006)). The present TIMN uses quite different approach to calculate the evaporation rates, as detailed in the text. In order to avoid confusion, this sentence has been deleted.

(3) Page 9, line 250: "We have extended the earlier QC studies of binary and ternary clusters to larger sizes."
Which sizes? Please indicate clearly which clusters are new, and which have been studied in previous publications. Also, list clearly the clusters for which QC data is applied (instead of other type of thermodynamic data). Are these the clusters listed in Tables 2-4?

Yes, these are clusters listed in Tables 2-4. The data from earlier studies and experimental data are properly marked, and notes describing their origin are given below the tables.

(4) Page 9, lines 254-255: The authors have used also a "locally developed sampling code, which creates a 'mesh' around the cluster, in which molecules being attached to the cluster are the mesh nodes", but this sentence is all that is said about the code. Please elaborate what this code exactly does, and give a reference, if possible.

The code is based on the following principle: mesh, with molecule to be added to the cluster placed in the mesh nodes, is created around the cluster, and blind search algorithm is used to generate the guess geometries. The mesh density and orientation of molecules are variable, as well as the minimum distance between molecules and cluster. We have added this elaboration to the revised manuscript.

(5) Page 10, lines 289-292: "Both the absolute values and trends in $\_G_{0+W}$ derived from calculations are in agreement with the laboratory measurements within the uncertainty range of $\_1$-2 kcal mol$_{-1}$ for both QC calculations and measurements. This confirms the efficiency and precision of QC methods in calculating thermodynamic data needed for the development of nucleation models."
$\_1$-2 kcal mol$_{-1}$ can be expected to be the general uncertainty of quantum chemical methods. However, as the Gibbs free energies are incorporated in the exponential factor of the evaporation rate (Eq. (7)), this uncertainty may propagate to an uncertainty of up to orders of magnitude in the particle formation rates and concentrations. This is discussed and demonstrated e.g. by Kürten at al. (2016), who estimated the uncertainties in the modeled particle formation rates by increasing or decreasing all Gibbs free energies by 1 kcal mol$_{-1}$. Depending on the conditions, this changes the formation rate by less than an order of magnitude, or even by up to several orders of magnitude. Please discuss also the sensitivity of the particle kinetics to the evaporation rates, and the impact of the uncertainties in $\_G$ on the formation rate.

We agree with the referee that the uncertainties in computed free energies of 1-2 kcal mol$^{-1}$ may lead to large uncertainties in particle formation rates under some conditions. However, uncertainties estimated by Kürten at al. (2016) represent the upper limit because estimates of Kürten at al. (2016) do not consider the error cancellation. In reality there probably does not exist such a thing as a systematic error of plus or minus 1 kcal mol$^{-1}$ assigned to each step of the cluster formation, because computed free energies may be overestimated for some clusters and underpredicted for others that leads to partial or, in some case, full error cancelation. In order to make it clear, we have added discussion on these matters in the revised manuscript.

(6) Page 11, lines 321-322: "Since positively charged $H_2SO_4$ dimers are expected to contain large number of water molecules, no quantum chemical data for these clusters are available."
What does this mean? Does it mean that the data cannot be computed at all, or that

the authors haven't computed such data in these studies?

This means that neither the authors nor other groups have computed these clusters. While it is hypothetically possible to compute them, no one has it done so far. Here the most difficult part is the adequate configurational sampling because the number of conformers is growing quickly with increasing hydration number and cluster size. We have modified the sentence to make it clear.

(7) Page 12, lines 348-351: "Table 3 shows that the presence of $NH_3$ in $H_2SO_4$ clusters suppress hydration and that $\_G_{0+W}$ for $S_2A_2$ falls below -2.0 kcal mol$^{-1}$. This is consistent with earlier studies by our group and others showing that large $S_nA_n$ clusters (n>2) are not hydrated under typical atmospheric conditions."
Please give references for these studies, especially for those conducted by other groups. Does this mean that all clusters and nanoparticles larger than $(H_2SO_4)_2(NH_3)_2$ are generally not hydrated, or do the particles become hydrated again at some larger size? At which size? What is assumed about the hydration of electrically neutral $H_2SO_4$-$NH_3$ clusters beyond the quantum chemistry data set, i.e. larger than $(H_2SO_4)_4(NH_3)_5$?
Generally, the hydration of a specific cluster ($S_2A_2$) tells nothing about the hydration of other clusters with different numbers of acid and base molecules. Therefore, it should be stated here that neglecting water in the larger clusters is just an assumption that has to be made due to the lack of thermodynamic data (as the authors have actually done later on line 454).

We have added references to the relevant studies in the revised manuscript:

Henschel, H., Navarro, J. C. A., Yli-Juuti, T., Kupiainen-Määttä, O., Olenius, T., Ortega, I. K., ... & Vehkamäki, H. (2014). Hydration of atmospherically relevant molecular clusters: Computational chemistry and classical thermodynamics. *The Journal of Physical Chemistry A*, *118*(14), 2599-2611.

Henschel, H., Kurtén, T., & Vehkamäki, H. (2016). Computational study on the effect of hydration on new particle formation in the sulfuric acid/ammonia and sulfuric acid/dimethylamine systems. *The Journal of Physical Chemistry A*, *120*(11), 1886-1896.

Herb, J., Nadykto, A. B., & Yu, F. (2011). Large ternary hydrogen-bonded pre-nucleation clusters in the Earth's atmosphere. *Chemical Physics Letters*, *518*, 7-14.

We agree with the Referee that the hydration of a specific cluster (S2A2) tells nothing about the hydration of other clusters with different numbers of acid and base molecules, and that neglecting water in the larger clusters is just an assumption that has to be made due to the lack of thermodynamic data. We have pointed this out in the revised manuscript. However, it is also important to note that the recent study by Henschel, H., Kurtén, T., & Vehkamäki, H. (2016) confirms our conclusion. In particular, Fig. 3 in their study shows clearly that in the case of fairly large cluster consisting of 3 $H_2SO_4$ and 3 $NH_3$ molecules, the average hydration number is less than 0.7 even if RH=100%.

(8) Page 12, lines 365-367: "This finding is fully consistent with the laboratory measurements showing that growth of neutral $S_sA_a$ clusters follows s = a pathway (Schobesberger et al., 2015)."
The study by Schobesberger et al. (2015) does not present any measurement data for neutral clusters. (Instead, they are modeled by the ACDC program in that study.)
Please correct the sentence.

Corrected.

(9) Page 13, lines 389-393: "_G0
+S values for S-S3–4Ww are consistently _1.5-3
kcal mol–1 less negative than the corresponding semi-experimental estimates (Table
4). The possible reasons behind the observed systematic difference are yet to be
identified and include the use of low-level ab initio HF method to compute reaction
enthalpies and uncertainties in experimental enthalpies in studies by Froyd and
Lovejoy (2003)."
The computed values for _G0+W (as well as for _G0+W for some positive clusters),
on the other hand, are more negative than those determined by Froyd and Lovejoy
(2003). Why doesn't the discussed systematic difference apply to these values?

Yes, it's applicable to all the relevant comparisons. Another important issue is that there exist
multiple sources of uncertainties in the data sets of Froyd and Lovejoy (2003). First of all, the data
sets for both positively and negatively charged clusters are not strictly experimental. While in the
case of negative clusters, the low level ab initio is used to get the final semi-experimental energy
values, in the case of positive ions, a theoretical thermochemical cycle is applied. The accuracy of
these "experimental" data are pretty much unknown; however, these data sets are currently the
only ones that report some sort of experimental values for negative and positive clusters of sulfuric
acid with water and, thus, we had no choice other than to compare computed data with these
particular data sets.

(10) Page 15, lines 471-472: "In the TIMN model, the equilibrium distributions are used
to calculate number concentrations weighted stepwise Gibbs free energy change"
Where is this averaged _G used in the model? Doesn't the model use averaged evaporation
frequencies (Eq. (10)), which are calculated over the individual evaporation
rates and thus do not correspond to the averaged _G?

The mode actually uses the averaged $\overline{\Delta G}_{s-1,s}$ to calculate averaged evaporation rates. To avoid
confusion, we have modified Eq. (10) (Eq. 12 in the revised manuscript).

(11) Page 15, lines 477-479: "In the atmosphere, where substantial nucleation is
observed, the sizes of critical clusters are generally small (s <_ 5−10) and nucleation
rates are largely controlled by the stability (or ) of small clusters with s <_ 5 − 10."
Please give references for this.

The number of $H_2SO_4$ molecules in critical clusters has been estimated from
d(lnJ )/d(ln[$H_2SO_4$]). A reference (Sipilä et al., Science, 2010) is now added.

(12) Table 2 and discussion on pages 17-19: For positively charged clusters, there is
QC data only up to clusters containing one $H_2SO_4$ molecule and two $NH_3$ molecules.
That is, not even the first growth step with respect to $H_2SO_4$ clustering (i.e. the
formation of a $H_2SO_4$ dimer) is covered, and in practice all the data for positive
clusters is guessed by using Eq. (11) (except for clusters containing more than 10 acid
molecules, starting from which data for electrically neutral bulk solutions is used also
for the positive clusters).
This is an extremely crude approximation. Please bring up this fact, and explain what
new "insights" we can learn about the thermodynamics of these clusters by using
these type of data.

We agree with the referee that the QC data for positively charged clusters are very limited and
the interpolation approximation is subject to large uncertainty. In order for the nucleation on

positive ions to occur, the first step is for $H_2SO_4$ to attach to a positive ion that does not contain $H_2SO_4$. Unlike negative ions, the effect of charge on the bonding of $H_2SO_4$ with positive ions is much weaker and thus the stepwise Gibbs free energy change for the addition of one $H_2SO_4$ molecule to form a positively charged cluster is likely to be similar to that of neutral clusters, i.e., decreasing with cluster size. Therefore, the QC data for positively charged clusters containing one $H_2SO_4$ molecule provides a critical constrain. The success of the model in predicting the [$NH_3$] needed for nucleation on positive ions to occur (Fig. 6) show the usefulness of the first step data and approximation. Nevertheless, we agree with the referee about the uncertainty and bring up the fact of the lack of thermodynamic data for positive ions in the revised manuscript.

(13) Page 17, lines 536-538: Similarly to positive clusters, the results for the thermodynamics of negative clusters raise some questions: "The effect of $NH_3$ on negative ions becomes important at s __ 4, when bonding between the clusters and $NH_3$ becomes strong enough to contaminate a large fraction of binary clusters with ammonia (Fig.3)." No QC data or experimental bulk data is used for the clusters around sizes s _ 4 − 9. Considering that this behavior is deduced by interpolating between QC data for small clusters that take up ammonia rather weakly, and macroscopic solution data for an electrically neutral $H_2SO_4$-$H_2O$-$NH_3$ liquid, it is difficult to see this result as very reliable. Please state that the thermodynamics of these clusters are highly uncertain (or explain why they would not be).

The interpolation is more than likely a reasonable approximation for negatively charged clusters with s=4-9, as indirectly confirmed by the success of our model in predicting the observed dependence of nucleation rates on [$H_2SO_4$] and [$NH_3$] (Figs. 6 and7). Please note that in many conditions where nucleation is significant, s<=~4 and the uncertainty associated with the interpolation is small. We agree that further experimental or QC study will be helpful to reduce the uncertainty and have empathized this in the revised manuscript.

(14) Table 4: The hydrate data for the negatively charged clusters is quite sparse for some clusters. Why are not hydrates with more water molecules considered for, for example, S–S1 or S–S2?

The hydration of these clusters is weak and, thus, does not impact the cluster formation because none of them are hydrated under typical atmospheric conditions (see refs. below).

Herb, J., Xu, Y., Yu, F., & Nadykto, A. B. (2012). Large hydrogen-bonded pre-nucleation $(HSO_4^-)(H_2SO_4)_m(H_2O)_k$ and $(HSO_4^-)(NH_3)(H_2SO_4)_m(H_2O)_k$ clusters in the Earth's atmosphere. The Journal of Physical Chemistry A, 117(1), 133-152.
Nadykto, A. B., Yu, F., & Herb, J. (2008). Towards understanding the sign preference in binary atmospheric nucleation. Physical Chemistry Chemical Physics, 10(47), 7073-7078.

(15) Tables 2-4: Why isn't all QC data that is used in the model given in the tables? For example, binary electrically neutral clusters are not included in Table 3. Please indicate clearly where these data can be found.
To keep the manuscript concise, we choose not to repeat results already published unless really necessary. We have provided references for the binary neutral clusters.
Nadykto, A. B., Al Natsheh, A., Yu, F., Mikkelsen, K. V., & Herb, J. (2008). Computational quantum chemistry: A new approach to atmospheric nucleation. Advances in quantum chemistry, 55, 449-478.

Herb, Jason, Alexey B. Nadykto, and Fangqun Yu. "Large ternary hydrogen-bonded pre-
    nucleation clusters in the Earth's atmosphere." Chemical Physics Letters 518 (2011): 7-14.

(16) Figure 3: Why are some hydrates with different numbers of water molecules
grouped together? For instance, $(H_2SO_4)_2(H_2O)_{1-7}$ is presented as one bar; this
doesn't tell much about the hydration as 1 and 7 are quite different numbers.
Also, in panel (d), please clarify that there is no hydrate data for these clusters; otherwise
the figure panel might be understood so that the clusters don't take up water at all.

We group together some clusters with different numbers of water molecules to make the figure
more clear and neat. We have clarified it in the figure caption for panel (d) as suggested by the
Reviewer.

(17) Figure 4: Why is the cumulative Gibbs free energy zero for the first growth steps
of the negative clusters in panel (b)? In panel (a), it does not look like these values
add up to zero, but should be negative instead.

$\overline{\Delta G}_{s-1,s}$ for small negatively charged clusters are strongly negative, implying that their formation
is barrierless. However, these small clusters cannot be considered as nucleated particles because
$\overline{\Delta G}_{s-1,s}$ can become positive for larger clusters due to the charge effect decreasing quickly as the
clusters are growing The negative $\overline{\Delta G}_{s-1,s}$ for small clusters is not able to cancel the positive
$\overline{\Delta G}_{s-1,s}$ for larger clusters and thus $\overline{\Delta G}_{s-1,s}$ for small clusters are set to zero when they are
negative in the cumulative Gibbs free energy calculation.

(18) Caption of Figure 4: "Calculations were carried out at T=292 K, RH=38%,
$[H_2SO_4]=3x10^8$ cm$^{-3}$ and $[NH_3]$= 0.3 ppb."
How were the vapor concentrations, e.g. $[H_2SO_4]$, used in the calculations?

P in Eq. (12) (Eq. 11 in the revised manuscript).

**Experimental bulk data for larger nanoparticles**
Bulk thermodynamic data is assumed for particles of all charging states containing at
least ten $H_2SO_4$ molecules. While this is in practice the only available option due to
the lack of other type of data, the approximation calls for some discussion about the
related uncertainties.
(19) At which conditions (temperature, partial pressures of the $H_2SO_4$, $NH_3$, $H_2O$
vapors) were the measurements (Marti et al., 1997; Hyvärinen et al., 2005) performed?
How reliably can it be extrapolated to different conditions outside the measurement
range?

The conditions of the measurements are given and possible uncertainties are discussed in the
revised manuscript.

(20) Page 16, lines 487-491: "Based on experimental data (Kebarle et al., 1967;
Davidson et al., 1977; Wlodek et al., 1980; Holland and Castleman, 1982; Froyd and
Lovejoy, 2003), the stepwise _G values for clusters decreases exponentially as the
cluster sizes increase and approaches to the bulk values when clusters containing
more than _8-10 molecules (Yu, 2005)."
Is possible size-dependent chemical composition, i.e. acid:base molar ratio, considered
here (e.g. Chen et al., 2018)? How does it affect the model results?

This is a general statement about the decrease of stepwise ΔG with the size of charged clusters. The possible size-dependent chemical position may be taken into account implicitly through the interpolation as the compositions of small clusters are different from those of large (s>10) clusters. Please see reply to comment #13 for the discussion concerning impacts of on our model results.

(21) Page 16, lines 491-494: "Cluster compositions measured with an atmospheric pressure interface time-of-flight (APi-TOF) mass spectrometer during CLOUD experiments also show that the chemical effect of charge-carrying becomes unimportant when the cluster contains more than 9 $H_2SO_4$ molecules (Schobesberger et al., 2015)." In the study by Schobesberger et al. (2015), it looks like the different charges approach similar composition somewhere in the size range where the $H_2SO_4$ content is _20-100 molecules (Figure 9 in the study). At 10 $H_2SO_4$ molecules, the composition of negative and positive particles is still different. Please comment.

Figure 9 of Schobesberger et al. (2015) shows that the difference in the composition of positively and negatively charged clusters quickly decreases as number of $H_2SO_4$ molecules increases from 1 to ~ 10 and exhibits little further changes. It is true that at 10 $H_2SO_4$ molecules, the composition of negative and positive particles is still different but the difference is much smaller than that in the case of small clusters. We have pointed this out in the revised manuscript.

(22) Page 17, line 524: "consistent with the laboratory measurements (Marti et al., 1997)" Isn't the discussed _G data derived from these measurements (i.e. naturally, it is consistent)? Please clarify.

Yes. The sentence has been modified to make it clear.

Page 24, lines 774-776: Is this the correct reference?

This is a wrong reference. Thank you for pointing this out. The correct one is:
Hyvärinen, A., T. Raatikainen, A. Laaksonen, Y. Viisanen, and H. Lihavainen, Surface tensions and densities of $H_2SO_4$ + $NH_3$ + water solutions, Geophy. Res. Lett., 32, L16806, doi:10.1029/2005GL023268, 2005.

**Approximated values for intermediate sizes with < 10 $H_2SO_4$ molecules**
(23) Eq. (11): What is this "extrapolation" formula based on? It is not clear why this functional form would be suitable for connecting QC and bulk measurements. Please explain clearly how the formula is derived, and discuss the related uncertainties.

Linear and exponential extrapolations are two common methods for this type of application. We choose exponential extrapolation as it fits better the stepwise ΔG change of neutral clusters that QC data are available. The related uncertainties are discussed in the revised manuscript.

(24) Page 16, lines 503-506: "c in Eq. (11) is the exponential coefficient that determines how fast $_G_{s-1,s}$ approaches to bulk values as s increases. In the present study, c is estimated from $_G_{s-1,s}$ at s=2 and s=3 for neutral binary and ternary cluster for which experimental (Hanson and Lovejoy, 2006; Kazil et al., 2007) or quantum-chemical data (Table 3) are available."
What can the data for clusters that contain 2 or 3 $H_2SO_4$ molecules possibly tell about how fast _G approaches bulk values?
Is c estimated based on QC data, experimental data, or both? How is this done exactly? Is it only for neutral clusters, or also for charged clusters?

It's an approximation. See our reply to comment 23 above.  In the present model, we estimated $c$ based on QC data of neutral clusters. We acknowledge that the extrapolation approximation is subject to uncertainty but this is the best approach we can come up with at this point in order to develop a model that can be applied to all conditions. Further QC and experimental studies of the thermodynamics of larger clusters can help to reduce the uncertainty.

(25) Finally, the most important issue regarding the thermodynamics is the fact that the "critical sizes", i.e. the barriers for nucleation, are located around cluster sizes for which there is no reliable thermodynamic data (Figure 4). For all different types of clusters (binary, ternary, all charging states), the maximum of the free energy curve is beyond the QC data (or just at the upper limit of the QC data in the case of negative ternary clusters). That is, the critical stage of nucleation is based on Eq. (11), which in turn does not seem to be based on an actual physical model.
Considering this, can the model really give important new information on $H_2SO_4$-$H_2O$-$NH_3$ particle formation mechanisms?

The maximum of the free energy curve shown in Fig. 4b is the accumulative free energy change and the maximum value (or nucleation barrier) is dominated by smaller clusters (Fig. 4a). In other words, the formation of small clusters are limiting steps and the uncertainty of stepwise $\Delta G$ for larger clusters where QC data are not available has limited impact on the predicted nucleation rate. As demonstrated in the paper, the model reveals the general favor of nucleation of negative ions, followed by nucleation on positive ions and neutral nucleation, for which higher $NH_3$ concentrations are needed, in excellent agreement with CLOUD measurements. The usefulness of the model can be seen from its success in reproducing the observed dependence of nucleation rates on various parameters and its ability to calculate nucleation rates under conditions for which measurements are not available.

**Kinetic model**
(26) In the kinetic model, the clusters are assumed to be in equilibrium with respect to both water and ammonia. Such equilibriation assumption can be made if the time scales of the attachment and evaporation processes of some compound are substantially shorter than those of other compounds. This is the case for water, as (a) its concentration (to which the attachment, i.e. molecular collision, frequency is directly proportional) is around _10 orders of magnitude higher than that of $H_2SO_4$ or $NH_3$, and (b) its binding to the clusters is so much weaker that its evaporation rate is _ several orders of magnitude higher than that of other compounds (except for some charged clusters in e.g. Table 2).
This is, however, generally not the case for ammonia. The binding of $NH_3$ depends strongly on the cluster composition: Depending on the acid:base ratio, either $NH_3$ or $H_2SO_4$ evaporates much faster than the other. Within the set of small clusters, the weakest and strongest bindings of $NH_3$ are of the same order as those of $H_2SO_4$ (e.g. Table 3). The collision rates of $NH_3$ are not necessarily multiple times higher than those of $H_2SO_4$, either: While ammonia is generally more abundant than $H_2SO_4$ in the atmosphere, there are environments where $[H_2SO_4]$ and $[NH_3]$ are around the same order (such as some of those simulated in this study).
Due to these reasons, the explanation for assuming equilibrium with respect to $NH_3$ is not justifiable (pages 13-14, lines 414-418): "In the lower troposphere, where most of the nucleation events were observed, $[H_2SO_4]$ is typically at sub-ppt to ppt level, while $[NH_3]$ is in the range of sub-ppb to ppb levels. This means that small ternary clusters can be considered to be in equilibrium with $H_2O$ and $NH_3$ vapors."
(a) Doesn't ammonia need to also evaporate much faster from the clusters for the equilibrium assumption to be justified? (b) At the simulated conditions, $[H_2SO_4]$ and $[NH_3]$ are in many cases of the same order. For instance, in Figure 6 at the lower end of the $[NH_3]$ axis, $[NH_3]$ is of the same order or even lower than $[H_2SO_4]$. In Figure

7a, [NH$_3$] is around 10 ppt, i.e. _ 10$_8$ cm$^{-3}$, and [H$_2$SO$_4$] is around 10$_7$. . . 10$_9$ cm$_{-3}$.

**Please show that the equilibrium with respect to NH$_3$ really is a valid assumption for these simulation systems.**

For the equilibrium assumption to be justified, the collision rate of clusters with NH$_3$ should be substantially higher than that with H$_2$SO$_4$. The evaporation rate of NH$_3$ depends on the composition of the cluster and can be very fast when NH$_3$:H$_2$SO$_4$ ratio are above one for small clusters. In many atmospheric conditions, especially in lower troposphere, [NH$_3$] is generally a few orders of magnitude higher than [H$_2$SO$_4$] and equilibrium assumption should be reasonable. For practical applications, nucleation rates are generally predicted based on the assumption that the clusters are in equilibrium and nucleation rates reach the steady state. Please note that the nucleation rates measured in CLOUD are also steady state values.

We agree with the Referee that the system may deviate from equilibrium if [NH$_3$] is less than or close to [H$_2$SO$_4$]. Under such cases, the equilibrium assumption may overestimate nucleation rates. We have added discussion on these matters in the revised manuscript.

(27) Some other aspects of the model also need clarification. The kinetic equations (Eqs.(1-6)) seem to include also collisions between charged clusters / particles of the same polarity. How high are the rate constants for such processes? Doesn't electrostatic repulsion prevent these attachments?
Further, if multiply charged particles can form in these collisions, how are these different charge numbers treated in the model? Shouldn't there be separate equations for particles that contain a single charge, two charges, three, and so on?

Yes, the electrostatic repulsion is too strong for small clusters to gain more than one charge. However, small charged clusters can be scavenged by large pre-existing particles of same polarity. Large pre-existing particles serve as the sink for small clusters in the model and the effect of multiple charge is small and thus is not tracked.

(28) Page 5, lines 162-166: "The initial negative ions, which are normally assumed to be NO$_{-3}$ , are converted into HSO$_{-4}$ core ions (i.e., S$_-$) and, then, to larger H$_2$SO$_4$ clusters in the presence of gaseous H$_2$SO$_4$. The initial positive ions H$_+$W$_w$ are converted into H$_+$A$_{1-2}$W$_w$ in the presence of NH$_3$, H$_+$S$_s$W$_w$ in the presence of H$_2$SO$_4$, or H$_+$A$_a$S$_s$W$_w$ in the case, when both NH$_3$ and H$_2$SO$_4$ are present in the nucleating vapors." What are the rate constants for the conversions of NO$_{-3}$ and H$_+$W$_w$?

This is a general statement of ion clustering process in the atmosphere when nucleation occurs. In the model we assume the starting negative ion is HSO$_4^-$. The rate constant for the conversion of initial positive ions to the one containing H$_2$SO$_4$ is ~ 2x10$^{-9}$ cm$^3$s$^{-1}$.

What does H$_+$A$_{1-2}$W$_w$ (or H$_+$S$_s$W$_w$ and H$_+$A$_a$S$_s$W$_w$) mean, i.e. how many ammonia and water molecules does it contain?
It's a general expression of cluster formula. As given in the Figure 1 caption, S, W, and A represent sulfuric acid (H$_2$SO$_4$), water (H$_2$O), and ammonia (NH$_3$) respectively, while *s*, *w*, and *a* refer to the number of S, W, and A molecules in the clusters/droplets, respectively.

In the equations (page 7, lines 192-193), "N$_{+,-}$ $_0$ and Q are the concentration of initial ions not containing H$_2$SO$_4$ and the ionization rate, respectively"
What do the "initial ions" refer to, e.g. H$_+$W$_w$ or H$_+$A$_{1-2}$W$_w$? NO$_{-3}$ or HSO$_{-4}$ ?

Initial positive ions include both $H^+W_w$ and $H^+A_{1-2}W_w$ (in equilibrium). Negative initial ion is $HSO_4^-$.

(29) Eq. (3): Why does the evaporation term for creating $H_2SO_4$ monomers from a dimer includes a factor of two (_j,2), but the corresponding collision term, removing monomers in the collision creating a dimer, does not?

Evaporation of one dimer generates two monomers. For the corresponding collision term (monomer with monomer), a factor of two (in loss) cancels the double count of collisions among monomers.

(30) Page 8, lines 212-213: "The methods for calculating _, , _, and _ for binary $H_2SO_4$-$H_2O$ clusters have been described in detail in Yu (2006b)."
I was not able to find the descriptions for _, _, and _ in the given reference; the paper only seems to re-direct the reader to discussion in 3 other papers. Please briefly summarize how these parameters are obtained.

We have added additional references and a brief description.

(31) Page 8, lines 221-222: "$N_o$ is the number concentration of $H_2SO_4$ at a given T under the reference vapor pressure P of 1 atm."
Isn't $N_o$ simply the number concentration corresponding to the reference pressure P of the QC calculations? What does it have to do with any [$H_2SO_4$]? In general, the evaporation rate should not be related to the concentration of any compound, as it does not depend on the composition of surrounding vapor (only on the temperature, i.e. the inert gas).

The referee is correct that the evaporation rate should not be related to the concentration of any compound. $N^o$ in the equation will be cancelled out with the $N^o$ in $\Delta G_{i-1,i}$. Details of the derivation and relationship can be found in the reference given (i.e., Yu, 2007). Please note that we have corrected a missed term in Eq. (8).

(32) Page 8, lines 223-225: "The temperature dependence of _$H_o$ and _$S_o$, which is generally small and typically negligible over the temperature range of interest, was not considered." Can you give a reference for the negligible temperature dependence?

The conclusion is based on typical calculated $\Delta C_p$, which largely controls the temperature dependence of $\Delta H$ and $\Delta S$ (see A.B. Nadykto et al. / Chemical Physics 360 (2009) 67–73 and references therein) and does not exceed a few tens of cal/mol/K in most cases studied here. The reference is added to the revised text.

(33) Page 19, lines 572-573: "mean evaporation rate ( ¯ ) of an $H_2SO_4$ molecule"
Is it assumed that only a single $H_2SO_4$ molecule evaporates, i.e. no water ligands, for instance, are attached to it? If so, please discuss the validity of this assumption, or even better, average the evaporation rates over all evaporation pathways with different numbers of other compounds attached to the acid molecule.

Yes, the present model assumes only a single $H_2SO_4$ molecule evaporates. This is likely the dominant evaporation pathway. We have pointed this out in the revised manuscript.

(34) Page 19, lines 573-574: "The shapes of ¯ curves are similar to those of _¯$G_{s-1,s}$ (Fig. 4a) as ¯ values are largely controlled by _¯$G_{s-1,s}$."
How is ¯ related to the averaged values _¯$G_{s-1,s}$? Isn't ¯ calculated based on individual values _$G_{s-1,s}$ (Eq. (10)), i.e. not exactly equivalent to _¯$G_{s-1,s}$?

Please see our reply to comment #10.

(35) The discussion on page 19, lines 575-584, feels somewhat confusing: First it's said that the effect of ammonia is significant for larger clusters and of less importance for small clusters (e.g. "the binding of NH3 to small neutral and charged clusters are weaker compared to that for larger clusters"), but after this it's concluded that "The nucleation rates, limited by formation of small clusters (s <_ 5), depend strongly on the stability or evaporation rate of these small clusters and, thus, on [NH3]."
So is or is not NH3 important for the small clusters and nucleation? Please clarify.

While the binding of NH3 to small neutral and charged clusters is weaker compared to that to larger clusters, small clusters containing NH3 are much more stable than those without (Fig. 4) and thus ammonia is important for nucleation. We have clarified this in the revised manuscript.

(36) Page 19, line 588: "the concentrations of clusters of all sizes are explicitly predicted"
A quasi-unary model cannot be called "explicit"; please re-formulate.

Please see our reply to the general comments in the beginning. To address the Reviewer's concern, we have deleted the word "explicitly" from the sentence.

(37) Eq. (13): Is it so that only growth through H2SO4 vapor is taken into account in the calculation of the particle formation rate? What about the effects of coagulation and recombination?
The quantity J that can be deduced from measurements -and that also is the relevant quantity for atmospheric modeling- includes all processes through which particles form, not only monomer condensation and evaporation. Therefore, these should be included also in the model-based formation rate.

For the chemical system considered in the present study, generally $N_1 \gg N_2 \gg N_3$ .... As a result, H2SO4 vapor growth dominates the steady state flux crossing 1.7 nm.

(38) Figure 1: The figure is confusing, and using patterns to fill the lines or spheres makes it somewhat difficult to read. For instance, it looks like "Condensation" means that electrically neutral clusters are ionized into charged particles (the arrows lead only to the charged blocks), and that "Coagulation / Scavenging" means that positively charged particles attached to each other or neutral particles. What is the difference between "Coagulation / Scavenging" and "Coagulation"?

We were trying to use "Scavenging" to represent the removal of small clusters by large pre-existing particles, also through coagulation. Condensation is actually implied in the green arrows. To avoid the confusion, we have deleted words "Condensation" and "Scavenging".

**Results and discussion**
(39) As a general comment, the description of the model should be a bit less ambitious. As one-compound discrete-sectional kinetic models have existed at least since the 1970s, the model cannot be considered "first", nor is it exactly "comprehensive" or "accurate" due to the quasi-unary assumption.
The addition of NH3 to the previous BIMN model does not make the model very new, either, as it means simply using different thermodynamic data in an existing model - and the main author has also previously published a modeling study entitled "Effect of ammonia on new particle formation: A kinetic H2SO4-H2O-NH3 nucleation model constrained by laboratory measurements" (Yu, 2006a). Besides, as the authors themselves also bring up, the kinetics of H2SO4-H2O-NH3 molecular clusters including the different charging states have been previously modeled e.g. by the ACDC program

(which the authors quite extensively criticize).

To address the Reviewer's concern, we have removed "first" and "accurate". For the reasons we gave in our reply to the general comments in the beginning, we think the present model is quite comprehensive.

(40) As previously (e.g. Nadykto et al., 2011; Nadykto et al., 2014), the main criticism is targeted at the modeling work by University of Helsinki (and this time also at the particle formation rate parameterization CLOUDpara based on the experimental data from the CLOUD chamber). In general, the authors criticize the ACDC model; however, the output of a clustering model is determined by the input parameters, namely the thermodynamic data. The ACDC program does not use any specific QC data, but the data is instead given by the user.
The ACDC data presented by Kürten et al. (2016) results from QC thermochemistry calculated with the RI-CC2/aug-cc-pV(T+d)Z//B3LYP/CBSB7 method. Therefore, the authors should call this rather e.g. "RI-CC2//B3LYP" data than "ACDC" data. The RI-CC2//B3LYP method is known to have a tendency to over-predict cluster stability, as has been discussed for example by the Helsinki group (e.g. Kupiainen-Määttä et al., 2015; Myllys et al., 2016), and thus it is not much used anymore in QC calculations.

The over-predictions of the thermochemical stability of nucleating clusters by RI-CC2//B3LYP used in ACDC code was actually first pointed out by Nadykto et al. (2014) and discussed  by Nadykto et al. (2015) and Kupiainen-Määttä et al. (2015). We agree with the Reviewer that ACDC program can use other types of QC data, however, the data obtained using ACDC we were referring to in the paper are based on RI-CC2//B3LYP thermochemistry.
In order to address the Reviewer's concern, we have replace "ACDC data" with "ACDC predictions based on nucleation thermochemistry obtained using RI-CC2//B3LYP method".

(41) Page 4, lines 122-123: "ACDC is also an acid–base reaction model, with the largest clusters containing 4-5 acid and 4-5 base molecules (no water molecules)":
This is not the case, as ACDC is simply a program that solves the kinetic equations (similar to Eqs. (1-6)) for a given set of molecular clusters using given thermodynamic input data, which does not need to involve acids or bases. It is not limited to some fixed specific largest cluster sizes; in the cited studies, the largest sizes were determined by the availability of QC data for the systems of interest.

We have deleted this sentence.

(42) Page 4, lines 127-130: "In ACDC, the nucleation rate is calculated as the rate of clusters growing larger than the upper bounds of the simulated system (i.e., clusters containing 4 or 5 $H_2SO_4$ molecules) (Kurten et al., 2016) and thus may over-predict nucleation rates when critical clusters contain more than 5 $H_2SO_4$ molecules."
It is of course not reasonable to model a system where the critical size region is outside the system boundaries. Thus, this region should be examined before simulating given conditions, as also discussed in the study by Olenius et al. (2013).

The second half of the sentence has been deleted.

(43) Page 4, lines 130-132: "All clusters simulated by the ACDC model do not contain $H_2O$ molecules and the effect of relative humidity (RH) on nucleation thermochemistry is neglected." Page 21, lines 645-646: "an important influence of RH on nucleation rates (which is neglected in both the CLOUDpara and ACDC models)"
The authors of the present manuscript are well aware of the fact that water can be included in the ACDC model: in fact, the effect of cluster hydration was recently the topic of a rather heated discussion between these authors and the researchers at University of Helsinki (Nadykto et al., 2014; Kupiainen-Määttä et al., 2015; Nadykto

et al., 2015; in this case, the question was about $H_2SO_4$-dimethylamine clusters), including i.a. ACDC simulations conducted as a function of RH.
Hydration can naturally be included in a kinetic model, such as ACDC, given that there is thermodynamic input data for clusters containing water. Please correct your claims about this. The effect of water in the $H_2SO_4$-$H_2O$-$NH_3$ system has been studied by ACDC e.g. by Henschel et al. (2016).

In view of the information the Reviewer provided us with, we have deleted this sentence.

(44) Also the particle formation rate parameterization by Dunne et al. (2016) is criticized. It would be fair to note that the deviations of the parameterization from the CLOUD data are not a new finding, as the uncertainties and weaknesses of the parameterization are discussed rather extensively in the work by Dunne et al. (e.g. supplementary Figures S3-S6).

We don't feel it is a criticism. We meant to point out the limitation of previous results which we aim to address in the present study.

(45) Page 11, line 333-334: "most of these studies, except for Nadykto and Yu (2007), did not consider the impact of $H_2O$ on cluster thermodynamics"
The effect of $H_2O$ on $H_2SO_4$-$NH_3$ clusters containing up to three $H_2SO_4$ and three $NH_3$ molecules has been considered by Henschel et al. (2014; 2016).

Thanks for the information. We have updated the discussion on this.

(46) Page 13, lines 396-397: The sentence "The binding of the second $NH_3$ to S–$S_3A$ to form S–$S_3A_2$ is much weaker than that of the first $NH_3$ molecule. This indicates that most of S–$A_a$ can only contain one $NH_3$ molecule" isn't clear: How does the binding of $NH_3$ to a cluster containing 3 $H_2SO_4$ molecules indicate something about the attachment of $NH_3$ to a bisulfate ion S–?

It's a typo. Should be $S^-S_3A_a$. Corrected.

(47) Comparisons to CLOUD data (Figures 6 and 7): Many of the comparisons look quite nice indeed. However, **more experimental data over a wider range of conditions should be shown to support the claim that the model is "in excellent agreement with CLOUD measurements"**.
For instance, in the work by Kürten et al. (2016) on CLOUD-based $J_{1.7}$, the model used in the study (ACDC with input thermodynamics computed with the RICC2//B3LYP method) is at some conditions in excellent agreement with CLOUD data, and at some conditions there are significant differences.
Therefore, comparisons with CLOUD data should be shown for **a large set of data**, for example the figures of the study by Kürten et al. (2016), including also electrically neutral cases and a wider range of ammonia concentrations.

We have extended the comparison with CLOUD data, including the neutral cases.

(48) Figure 6: The original CLOUD data includes also $J_{1.7}$ for experiments with no ions. Please add these electrically neutral experimental and model data to the figure. It looks like the slope of the modeled $J_{1.7}$ is quite steep when neutral nucleation takes over; it is interesting to see how this compares with the measurements.

Neutral cases without ions are now included and discussed.

(49) Figure 7, top panel: For most lines, there are only 3 experimental data points, which doesn't make the comparison of these data to the model lines very strong. As

there is so much CLOUD data available, please pick more representative data from e.g. the work by Kürten et al. (2016).
Especially low but still non-negligible ammonia mixing ratios are not shown in the current comparisons. If the model is said to cover "a wide range of atmospheric conditions", these should be included.

To be comparable, [$NH_3$] and T should be the same for each line, which limits the number of experimental points. We have extended the comparison with CLOUD data in separate figures.

Technical comments:
(50) Change all occurrences of "physio-chemical" to "physico-chemical"; presumably "physio" refers to physiology, not physics.

Done. Thanks for pointing these out.

(51) Page 2, line 35: Change "specie" to "species".

Done.

(52) Page 9, lines 240-245: The sentence "In earlier studies, this method has been applied to a large variety of atmospherically-relevant clusters and has been shown to be well suited to study the ones, (...)" is clumsy (i.e. what does "the ones" refer to?); please re-formulate.

Changed "the ones" to "the $H_2SO_4$-$H_2O$ and $H_2SO_4$-$H_2O$-$NH_3$ clusters".

(53) Page 9, line 253: Change "basin hoping" to "basin hopping".

Done.

(54) Page 11, line 332: It is misleading to list Kürten et al. (2015) as a computational study, as it doesn't present any computationally obtained thermodynamics.

We have changed Kürten et al. (2015) to Kürten et al. (2007) and added it in the reference list.

(55) Page 16, line 505: Change "cluster" to "clusters".

Done.

(56) Table 1: Please give units for the energy quantities. Please also clarify that "H" and "S" may refer to either the energetics, or the cluster composition (the first column), or use different symbols for some of the abbreviations / quantities. Also change "based" in the footnote to "based on".

That's a good point. Instead of using abbreviations, we keep the original words in the table. Units are now given in the table.

"based" in the footnote of Table 2 has been changed to "based on".

(57) The resolution and/or clarity of some figures, mainly 1 and 3, is rather poor. Please fix this.

Fixed.

---

## Author Response (AR2)

Dear Dr. Kerminen,

Thank you for obtaining the review comments of our revised manuscript. We have addressed all the comments and revised the manuscript accordingly, as detailed below. All changes made to the manuscript have been marked with Track-Change tool in one of submitted files.

Please let me know if there are any questions.

Best regards,

Fangqun Yu

Comments on the revised manuscript by Yu et al.
The authors thank the referee again for taking time to review the revised manuscript and provide constructive comments, which have allowed us to further clarify and improve the manuscript. Our point-to-point replies to the comments are given below, with the original comments in black, and our response in blue.

I'm happy to see that the authors have included more data in the measurement comparisons, and also made the discussion more balanced. I have one main suggestion regarding the new comparison (Figure 8):
As discussed, the agreement between CLOUD measurements and the presented model becomes worse when (1) temperature increases, and (2) ions are not present. It would be good to note that these are the conditions when the role of cluster evaporation (i.e. thermodynamics) becomes more important (i.e. higher evaporation and/or generally less tightly bound clusters), and thus they are likely to reveal the biases of the used thermochemistry. This naturally applies to all thermodynamic data sets, PW91PW91/6-311++G(3df,3pd) and other, regardless of the kinetic model framework used.
Agree. We have pointed this out in the revised manuscript.

Also, will the model be freely available?
The model is presently not yet in the public domain. We will continue to evaluate and improve the model by comparing with more measurements (including those taken in the atmosphere). On the other hand, we will make the parameterization of TIMN based on the present model available to the community (in term of lookup tables) after the publication of this manuscript. The TIMN parameterization can be easily used in 3-D models to calculate TIMN rates under a wide range of atmospheric conditions.

In addition, some of the replies to my previous comments were slightly inadequate. For instance, the following points would still need a bit of elaboration:

Comment on the model being quasi-unary: The authors reply that "the model is multicomponent" - I agree that the thermodynamic data is multi-component, but the kinetic model is not. The kinetic equations consider only the number of $H_2SO_4$ molecules i in each particle (page 7): "$N_i$ is the total number concentration (cm-3) of all cluster/particles (binary +

ternary) in the bin i. For small clusters (i≤id), Ni is the number concentration (cm-3) of all clusters containing i H2SO4 molecules." To the best of my knowledge, this means that the model is quasi-unary, and also the authors have used the same term for the previous versions of the model (Yu, J. Chem. Phys., 127, 054301, 2007). If the model was explicitly multi-component, then there would be no reason to apply the equilibrium assumptions for e.g. ammonia.

We respect this different perspective with regard to quasi-unary. It does not affect the findings and conclusions of the present work.

Comment 1: I did not ask about the QC data, I asked if the different approaches to assess the thermochemistry of clusters and particles of different sizes, compositions and charging states (QC, Eq. (10), experimental liquid data…) could be presented in an easy-to-read way. How about including this information e.g. in Figure 1? That is, explain for each charging state which size range is described with which thermochemical data; it would be much easier than digging the information from the text.

Our fault for the mis-interpretation of the comment. Actually this information can be readily found in Fig. 4a where the clusters using QC data are marked with symbols and those based on Eq (10) and experimental liquid data are described in Figure caption. To mark "QC, Eq. (10), experimental liquid data" in Figure 1 will make Figure 1 too busy. Therefore we didn't modify Figure 1.

Comment 17: I don't understand why you artificially set the cumulative Gibbs free energy to zero when it should be negative. Free energy profiles exhibiting both a minimum and a maximum can naturally occur, especially for charged clusters (see e.g. Figure 2 in Vehkamäki and Riipinen, Chem. Soc. Rev. 41, 5160-5173, 2012), and it simply means that there exist stable "pre-nucleation" clusters.

Got the point. To address the referee's concern, we have updated Fig. 4 for negative ions to include the negative deltaG in accumulative Gibbs free energy. The change is just for the approach to present and does not affect the TIMN rates. The related text is also updated accordingly.

Comment 18: The vapor concentrations were probably also used to convert the QC data to the given conditions (through the law of mass action)?

Yes.

Comment 24: I understand that parameter c is an approximation, but the statement "We estimated c based on QC data" still does not answer the question about how c is exactly calculated.

As stated in the text (Lines 388-391), "In the present study, $c$ is estimated from $\Delta G_{s-1,s}$ at $s=2$ and $s=3$ for neutral binary and ternary clusters for which experimental (Hanson and Lovejoy, 2006; Kazil et al., 2007) or quantum-chemical data (Table A3) are available". We feel that this explains how c is calculated. We slightly modified the sentence to make it clearer.

"In the present study, $c$ is estimated by fitting $\Delta G_{s-1,s}$ at $s=2$ and $s=3$ based on Eq. (10) to those from experimental (Hanson and Lovejoy, 2006; Kazil et al., 2007) or quantum-chemical data (Table A3)."

Comment 25: I fully agree that "the formation of small clusters are limiting steps", but the particle formation process is limited by cluster stability throughout the size range where the clusters are not stable, which is at least up to the barrier maximum. This is clear e.g. from simplified kinetic models such as classical nucleation theory, where the cluster free energy at the maximum of the barrier (the "critical cluster") is the only free energy determining the particle formation rate. I am sure the authors agree, since they have recently used such critical-cluster-based approaches themselves (Yu et al., Atmos. Chem. Phys. 17, 4997-5005, 2017).

Also, repeating that the model is "in excellent agreement with CLOUD measurements" is not helpful, when Figure 8 tells that this is not the case in all conditions. Thus, there is no need to commend a model when it is not justifiable, but it is instead good to point out the weaker features (as the authors already have nicely done in replies to some other comments).

We agree with the referee that "the particle formation process is limited by cluster stability throughout the size range where the clusters are not stable". When we say that "the formation of small clusters are limiting steps", we mean that the cluster free energy at the maximum of the barrier (the "critical cluster") is dominated by small clusters.

We searched the whole manuscript, only in one place (line 22) we found "in excellent agreement with CLOUD measurements". However, this statement refers to "The model reveals the general favor of nucleation of negative ions, followed by nucleation on positive ions and neutral nucleation, for which higher NH3 concentrations are needed," (Lines 20-22). Therefore, this statement is justified.

We agree that Figure 8 shows the difference between model prediction and CLOUD measurements for neutral nucleation at high temperature. This has been pointed out in the text, along with possible reasons. It should be noted that in the real atmosphere ionization is always present and under such a condition TIMN model is overall in excellent agreement with CLOUD measurements.

Comment 26: It might be good to clearly note then, that the model scheme is probably not suitable for situations where ammonia concentration is not substantially higher than H2SO4 concentration.

To address the referee's concern, the sentence (Line 304) has been modified to clearly note this.

To amend a few issues:
"Please note that the nucleation rates measured in CLOUD are also steady state values": Equilibrium and steady state are two different things, and the fact that CLOUD formation rates are assessed for a steady state has nothing to do with the assumption regarding cluster equilibration with respect to ammonia. (Equilibrium is a steady state with no net formation or growth of particles. A steady state where particle formation occurs is not equilibrium, but instead any time-independent situation with or without cluster equilibration with respect to some chemical compound.)

Got the point.

"It should be noted that all previous ternary nucleation models discussed in Section 2.1 assume the equilibrium with respect to NH3": No, they actually don't. For instance, the acid-base scheme used by Chen et al. (Proc. Nat. Acad. Sci., 109, 18713-18718, 2012), and further developed by Jen et al. (J. Geophys. Res. Atmos., 119, 7502-7514, 2014), assumes two separate acid dimers (clusters containing two acid molecules) that have different base content. ACDC (McGrath et al., Atmos. Chem. Phys., 12, 2345-2355, 2012) does not make any equilibrium assumptions with respect to ammonia.

We meant all previous CLASSICAL ternary nucleation models discussed in Section 2.1., Coffman and Hegg, 1995; Korhonen et al., 1999; Napari et al., 2002.

Got the referee's point with regard to the equilibrium assumption. No change is needed for the text as the above comments refer to the discussion in our previous reply to the referee's comments.

Comment 27: Please add these clarifications also to the manuscript (I was not able to find them there).
Yes. These clarifications are now included in the revised manuscript.

Comment 28: If Eqs. (1) and (2) correspond to H+AaWw and NO3-, in which equation is the bisulfate ion HSO4- included, i.e. does index i in Eq. (5) refer to the sum of H2SO4 and HSO4-? What does the second term of Eq. (2) describe; is NO3- evaporating from a negative cluster or HSO4-?
Yes, index i in Eq. (5) refers to the sum of H2SO4 and HSO4-. The second term of Eq. (2) describes the reaction of HSO4- + HNO3 $\rightarrow$ NO3- + H2SO4. Although the rate of this reaction is generally negligible, we keep the term there for completeness. We have clarified these in the text (Lines 194-196).

Comment 29: Yes, but isn't the double count canceled also for the evaporation term? This is because the evaporation rate constant gamma *includes* the collision rate constant beta as given by Eq. (7), and the permutation factor of 1/2 should be included in beta. Or is the beta in Eq. (7) defined differently than the beta in Eq. (3)?

The beta in Eq. (7) is the same as the beta in Eq. (3). In our definition, beta, as defined in reference cited (Yu, 2007), is simply the collision rate constant and does not contain the permutation factor of 1/2. Therefore, Eq. (3) is correct.

Comment 31: There's something wrong with the updated Eqs. (7) and (8), since the H2SO4 concentration N1,0 doesn't cancel out; please fix this. In any case, there is no reason to include any H2SO4 concentrations in the equation of evaporation rate, as they are not needed there. The original comment was mainly related to the statement "N0 is the number concentration of H2SO4 at a given T under the reference vapor pressure P of 1 atm". In QC methods, N0 is the arbitrary number concentration of a hypothetical gas consisting solely of the species for which the calculation is performed (which can be a single molecule or a cluster), and doesn't have to do with any concrete H2SO4 or other vapor concentration. Conversions are not needed, as they cancel out in the evaporation rate anyway, as the authors state.

The referee is correct. We have changed Eq. (8) back to the original one (as in ACPD) and pointed out that N0 is  the arbitrary number concentration of a hypothetical gas consisting solely of the species for which the calculation is performed (generally under the reference vapor pressure P of 1 atm).

Comment 43: It is still claimed in Section 3 of the revised manuscript (page 18, line 554) that the ACDC model neglects the effect of water.

[revised manuscript text omitted]